# Interactive 3-D visual analysis of ERA5 data: improving diagnostic indices for marine cold air outbreaks and polar lows

Marcel Meyer[1,3], Iuliia Polkova[2,3], Kameswar Rao Modali[1], Laura Schaffer[2], Johanna Baehr[2,3], Stephan Olbrich[1,3], and Marc Rautenhaus[1,3]

[1]Regional Computing Centre, Visual Data Analysis Group, Universität Hamburg, Germany
[2]Institute of Oceanography, Universität Hamburg, Germany
[3]Center for Earth System Research and Sustainability (CEN), Universität Hamburg, Germany

**Correspondence:** Marcel Meyer (marcel.meyer@uni-hamburg.de)

**Abstract.** Recent advances in visual data analysis are well suited to gain insights into dynamical processes in the atmosphere. We apply novel methods for three-dimensional (3-D) interactive visual data analysis to investigate Marine Cold Air Outbreaks (MCAOs) and Polar Lows (PLs) in the recently released ERA5 reanalysis data. Our study aims at revealing 3-D perspectives on MCAOs and PLs in ERA5 and at improving the diagnostic indices to capture these weather events in long-term assessments on seasonal and climatological time-scales. Using an extended version of the open-source visualization framework *Met.3D*, we explore 3-D perspectives on the structure and dynamics of MCAOs and PLs and relate these to previously used diagnostic indices. Motivated by the 3-D visual analysis of selected MCAO and PL cases, we conceptualize alternative index variants that capture the vertical extent of MCAOs and its distance to the dynamical tropopause. The new index variants are evaluated, along with previously used indices, with a focus on their skill as a proxy for the occurrence of PLs. Testing the association of diagnostic indices with observed PLs in the Barents and the Nordic Seas for years 2002-2011 shows that the new index variants based on the vertical structure of cold air masses are more skillful in distinguishing the times and locations of PLs, compared with conventional indices based on sea-air temperature difference only. We thus propose using the new diagnostics for further analyses in climate predictions and climatological studies. The methods for visual data analysis applied here are available as an open-source tool and can be used generically for interactive 3-D visual analysis of atmospheric processes in ERA5 and other gridded meteorological data.

## 1 Introduction

Marine Cold Air Outbreaks (MCAOs) are transport events of cold air from sea-ice or snow-covered regions towards relatively warmer oceans (Rasmussen, 1983; Kolstad and Bracegirdle, 2008; Gryschka, 2018). Understanding MCAOs is relevant because they represent conditions favourable for extreme weather phenomena known as Polar Lows (PLs; Rasmussen, 1983; Ese et al., 1988; Kolstad, 2011; Michel et al., 2018). PLs are intense mesoscale cyclones, which have been called "Arctic hurricanes" (Nordeng, 1992; Føre et al., 2012; Bracegirdle, 2012) due to similarities with tropical hurricanes including symmetric vortex-like cloud patterns. PLs in the Northern Hemisphere usually occur during winter and are characterized by strong winds, heavy precipitation and severe marine icing, which pose substantial risks to marine activities and infrastructures (Aarnes et al.,

2018). Improving the understanding of MCAOs and their relation to PLs could improve marine services in the polar regions. In addition to representing conditions conducive for extreme weather, MCAOs are also important in the context of deep water formation as they contribute to cooling of the ocean surface (Papritz and Spengler, 2017).

To facilitate statistical statements about atmospheric processes in, e.g., climatological or in climate prediction studies, weather phenomena are commonly characterized by means of diagnostic indices that abstract a complex atmospheric structure into a simple numerical value. To define such indices for MCAOs and PLs, various interdependent factors need to be considered, and understanding the link between MCAOs and PLs remains an area of active research (Claud et al., 2007; Kolstad and Bracegirdle, 2008; Kolstad, 2011; Terpstra et al., 2016; Afargan-Gerstman et al., 2020; Stoll et al., 2021). To diagnose MCAOs, most previous studies used an index that represents a simplified version of the Brunt-Väisälä frequency for quantifying static stability, by considering only the sign of the vertical temperature gradient between potential skin temperature of the ocean and potential air temperature aloft (e.g., Papritz et al., 2015; Kolstad, 2017). The temperature gradient is termed MCAO index, where positive potential temperature difference between the ocean surface and the air aloft indicates vertical instability. Previously used MCAO indices (Papritz et al., 2015; Kolstad, 2017) have the form

$$m_\theta = \theta_{\text{skin}} - \theta_{\text{850hPa}}, \tag{1}$$

or variations thereof (Kolstad et al., 2009; Fletcher et al., 2016; Papritz and Sodemann, 2018; Landgren et al., 2019), where $\theta_{\text{skin}}$ is the potential skin temperature (which is the sea surface potential temperature), and $\theta_{\text{850hPa}}$ is the potential air temperature at 850 hPa. Other related indices use the difference between the temperature at a certain height and the sea surface temperature (e.g., Zappa et al., 2014). In what follows, we will summarize the different variants of previously used MCAO indices using the term *conventional MCAO index* (to distinguish these from other metrics considered here). The vertical level at which air aloft is considered for calculation of the conventional MCAO index (850 hPa in Eq. 1) will be referred to as the *characteristic pressure level*.

The choice of the characteristic pressure level used for computing the conventional MCAO index varies substantially amongst previous studies - from 500 hPa (Landgren et al., 2019), to 700 hPa (Kolstad et al., 2009), 800 hPa (Fletcher et al., 2016), 850 hPa (Papritz et al., 2015; Kolstad, 2017; Polkova et al., 2019, 2021), to 900 hPa (Papritz and Sodemann, 2018). Consequently, the magnitude of the MCAO index values, as well as the geographical extent of areas with positive MCAO index values differ between studies, and a classification into weak, moderate and strong MCAO events requires adjustment accordingly. For small variations of the characteristic pressure level, the effect on index values can be small if the pressure level is located inside of a particularly well-mixed layer with a fairly uniform vertical profile of potential air temperature. However, large variations of the characteristic pressure level can have a substantial effect on index values. On the one hand, the variety of different pressure levels considered in different previously used MCAO indices provide a diverse and valuable knowledge base, but on the other hand it also complicates comparison and interpretation of results from different studies. Open questions remain on the appropriate choice of a characteristic pressure level and the sensitivity of results to this choice (e.g., the frequency of occurrence

of MCAOs in climatological assessments). Since MCAOs have been reported to be a necessary, but not a sufficient condition for the occurrence of PLs (Kolstad, 2011), an open question is whether a diagnosed presence of MCAOs can be useful as a proxy for the occurrence of PLs. A recent climatological study finds a weak relation between MCAOs and the occurrence of polar mesoscale cyclones, including PLs (Michel et al., 2018). In this study, we investigate the effect of the choice of the characteristic pressure level for the MCAO index and the vertical structure of MCAOs in relation to PLs. We develop and test alternative indices that do not rely on a subjective element in the choice of a characteristic pressure level, but instead take into account the vertical structure of MCAOs and PLs. We quantify the link between MCAO indices and the time and location of observed PLs and propose a simple method for determining the characteristic pressure level that maximizes this link.

For our analyses we make use of recently released reanalysis data and innovative methods for 3-dimensional (3-D) interactive visual analysis (IVA). The global reanalysis dataset ERA5 (in the following referred to as ERA5), recently released by the European Centre for Medium-Range Weather Forecast (ECMWF), is considered to be the most detailed and highest quality global reanalysis data available (Hersbach et al., 2020; Copernicus Climate Change Service (C3S), 2017). The increased spatial and temporal resolution of ERA5, compared to its predecessors, along with its large temporal coverage, allows for both, 3-D analysis of mesoscale atmospheric phenomena during single weather events and statistical analysis over multiple decades. Recent advances in graphics hardware along with innovative methods for visual data analysis facilitate the interactive 3-D visual exploration of ERA5 and other gridded meteorological data. Rautenhaus et al. (2018) provide a comprehensive overview of methods and potential benefits. Examples of 3-D IVA include supercell tornados (Orf et al., 2017), jet-stream core lines (Kern et al., 2018), and synoptic-scale fronts (Kern et al., 2019). 3-D IVA provides more comprehensive impressions and can help improving our understanding of meteorological phenomena, for example by rapid visual investigation of dynamical processes and by exploration of unknown features for formulating new hypotheses. The potential of 3-D depictions has long been appreciated by meteorologists (see e.g., Uccellini, 1990; Rautenhaus et al., 2018). However, despite novel 3-D visualization being available in several frameworks (open-source examples include *ParaView* (Ayachit, 2015), *Vapor* (Li et al., 2019) and *Met.3D* (Rautenhaus et al., 2015b,a), meteorological studies are still mainly conducted by means of visualizing static 2-D slices. Rautenhaus et al. (2018) discuss reasons for the slow uptake of modern visualization methods in the atmospheric sciences, including usability aspects as well as too few studies in the atmospheric community that demonstrate the potential benefits to be gained from, e.g., 3-D IVA. In this study, we extend and apply the open-source interactive visualization framework *Met.3D* (Rautenhaus et al., 2015b,a) for interactive 3-D investigation of the structure of MCAOs and PLs as represented in ERA5. We investigate to which level of detail the 3-D structure of MCAOs and PLs is resolved in ERA5, demonstrate the potential of 3-D IVA as a tool to understand atmospheric processes, and use insights from 3-D IVA as inspiration for conceptualizing improvements to existing MCAO and PL indices.

The objectives for this study are (a) to obtain a 3-D perspective on the structure and dynamics of MCAOs and PLs, (b) to relate the 3-D structure to previously used diagnostic indices for representing these weather events on seasonal and climatological time-scales, and (c) to evaluate diagnostic indices in the context of observed PLs for the main purpose of testing if these indices

could serve as proxies for PLs. The article is structured as follows. Sect. 2 describes the data, the visual analysis setup, the candidates for improved indices and the methodology to evaluate these. In Sect. 3 we illustrate insights gained from the 3-D visual analysis and describe the evaluation of diagnostic indices. The article is concluded in Sect. 4.

## 2  Data and Methods

Our analysis starts with the interactive 3-D visual exploration of selected cases of MCAOs and PLs in ERA5 (Sect. 2.1 and 2.2). Subsequently, we conceptualize alternative diagnostic indices (Sect. 2.3) using insights from the 3-D IVA as inspiration. The performance of the new indices is tested in comparison with conventional MCAO indices by assessing associations with observed PLs (Sect. 2.4).

### 2.1  ERA5 data

Our analyses are based on the recently released ECMWF ERA5 reanalysis, with a spatial resolution of approximately 31 km horizontally, 137 vertical model levels, and hourly temporal resolution (Hersbach et al., 2020; Copernicus Climate Change Service (C3S), 2017). We analyse data covering the time-interval 2002-2011, which is chosen to cover the times of observed PLs in the STARS data (Sect. 2.4). For use with Met.3D, data is regridded to a regular longitude-latitude grid (0.25° horizontal resolution) and to a polar stereographic grid (25 km horizontal resolution). In the vertical, all model levels are retained; if necessary, vertical interpolation between model levels is conducted on-the-fly by Met.3D.

### 2.2  Interactive visual analysis of ERA5 with Met.3D

Met.3D is a meteorology-specific 3-D visualization framework that provides the user with various methods for IVA of gridded meteorological data (Rautenhaus et al., 2015b,a). The framework focuses on interactive rapid exploration of the 3-D atmosphere and on uncertainty represented by ensemble simulations. It is designed to bridge the gap between 2-D visualizations (including horizontal maps, vertical cross-sections, vertical profiles), 3-D visualizations (including isosurfaces, direct volume rendering, 3-D streamlines and trajectories) and novel feature-based displays (Kern et al., 2018, 2019). All visualization techniques can be combined in an interactive setup, so that benefit can be gained from augmenting, e.g., traditional 2-D map views by corresponding 3-D perspectives. As part of this study, we update Met.3D by extending it with implementations of two new features with relevance to the analysis of MCAOs and PLs in ERA5 data: (i) support for visualizing polar stereographic data, as well as other common map projections; (ii) on-the-fly computation of MCAO indices on user-defined vertical levels for visual assessment of the sensitivity to the choice of the characteristic pressure level in conventional MCAO indices. The current release (v1.7) of Met.3D can be used generically for 3-D IVA of atmospheric phenomena in ERA5 (Met.3D - Homepage, 2021; Met.3D - Documentation, 2021; Met.3D - Documentation ERA5, 2021) and other gridded meteorological datasets.

We apply Met.3D to investigate the 3-D structure of MCAOs and PLs in ERA5. For the initial explorative phase of the investigation, a set of MCAO and PL cases were selected based on previous studies (Kolstad, 2011; Føre et al., 2012; Bracegirdle,

2012) about strong MCAO events and symmetric PLs (e.g., the case on Dec. 18, 2002 in Fig. 1 and 2). After the statisti-
cal evaluation of new index variants (Sect. 2.4), we conducted additional interactive visual analyses for one exemplar event
characterized by an overlap of PL occurrence with high values of the new MCAO index (the case on Mar. 24, 2011 in Fig. 1
and 2). The following exploratory visualization methods were used to study the ERA5 atmosphere: interactive sliding of 2-D
horizontal and vertical cross-sections through the 3-D atmosphere, exploration of the shape and location of 3-D isosurfaces
of selected variables including potential temperature and wind speed, direct volume rendering to inspect cloud liquid and ice
water, computation of 3-D streamlines of wind fields. During the initial phase of the visual case-analyses, we explored >10
ERA5 variables over the Northern Hemisphere (north of 25° N) for several cases of MCAOs and PLs to obtain a picture of
the large-scale atmospheric situation. During the second phase, we visually analysed single cases of MCAOs and PLs in more
detail by inspecting ERA5 variables (*t, pv, u, v, w, z, q, cc, ciwc, clwc*) on a smaller domain covering the Barents and the Nordic
Seas. More technical details are given in Appendix A.

## 2.3   New indices for MCAOs and PLs

Conventional MCAO indices have been calculated with a variety of different characteristic pressure levels, ranging from 500
to 900 hPa (Landgren et al., 2019; Kolstad et al., 2009; Fletcher et al., 2016; Papritz et al., 2015; Kolstad, 2017; Papritz and
Sodemann, 2018; Polkova et al., 2019). For understanding the 3-D structure of MCAOs and the sensitivity of conventional
MCAO indices to the choice of the characteristic pressure level, we have implemented a functionality in Met.3D that allows
for IVA of the effect of varying the characteristic pressure level. For this purpose, we introduce a simple 3-D extension of
conventional MCAO indices. We compute the temperature difference between the surface and each vertical pressure level, $p$,
instead of considering only one particular characteristic pressure level, as was usually done in previous studies on MCAOs.
That is, instead of the conventional MCAO index (Eq. 1), we compute its 3-D variant,

$$m_\theta^p = \theta_{\text{skin}} - \theta_p, \tag{2}$$

and use methods for interactive visual data exploration for its analysis (e.g. sliding a horizontal cross-section through all vertical
levels, $p$, of $m_\theta^p$). In a similar way, we implement in Met.3D the conventional MCAO index variants described in Kolstad and
Bracegirdle (2008), Landgren et al. (2019), and Fletcher et al. (2016). Insights from the 3-D IVA (Sect. 3.1 and 3.2) are used
as inspiration for conceptualizing improved diagnostic indices for capturing MCAOs and PLs. We summarize the definition of
indices here and describe the results from the IVA in the results section. All indices used in this study are summarized in Table 1.

Two new diagnostic indices are introduced and tested: (i) the new MCAO index, and (ii) the new PL index. Put simply, the new
MCAO index approximates the vertical extent (in what follows also termed "height") of MCAOs, and the new PL index mea-
sures the vertical distance between the upper boundary of MCAOs and the dynamical tropopause. These metrics are designed
to address shortcomings in conventional indices (e.g., a subjective choice of pressure level and weak relation between MCAOs
and PLs) by taking into account 3-D features of the atmospheric circulation, while remaining simple and computationally

cheap, and hence feasible for use in further climatological studies that rely on processing of large amounts of data.

(i) The new MCAO index approximates the vertical extent of the lower-level instability induced by MCAOs (expressed as a pressure difference). It is calculated for each horizontal grid-cell and time-step (hourly in ERA5) as the pressure difference between the surface and the upper boundary of the lower-level instability caused by a MCAO,

$$m_p = p_0 - p^*, \tag{3}$$

where $p_0$ is defined here as the standard constant surface pressure 1013.25 hPa. For the computation of the upper boundary of the MCAO, $p^*$, we determine the pressure level at which the potential air temperature equals the potential skin temperature. The new MCAO index, $m_p$, is set to 0 in grid-cells without lower-level instability ($\theta_{skin} < \theta_p$ for the vertical levels above each grid-cell). High index values of $m_p$ correspond to high vertical extents of the instability region caused by MCAOs.

(ii) The new polar low index measures the vertical distance between the upper boundary of MCAOs and the dynamical tropopause. It is calculated for each horizontal grid-cell and time-step within the area of MCAOs ($m_p > 0$) as the pressure difference

$$m_{tr} = p^* - p_{tr}, \tag{4}$$

where $p_{tr}$ is the pressure at the dynamical tropopause, which is defined by the 2 Potential Vorticity Unit (PVU) surface. The new PL index decreases with decreasing distance between the dynamical tropopause and the upper boundary of MCAOs and turns negative if the lower-level instability extends all the way to the dynamical tropopause and above ($p^* < p_{tr}$).

## 2.4 Evaluating the link between indices and observed PLs

The conventional and the new indices are evaluated by comparison with observed PLs, as reported in the STARS dataset (polar low tracks north, STARS - data, 2013; Noer et al., 2011). The STARS dataset (hereafter also referred to as STARS) contains 140 PLs observed during years 2002-2011 for the geographical region of the Barents and the Nordic Seas, including data about the times of PLs, the PL track and the approximate PL radius. For comparing diagnostic indicators with observed PL data, we use a 2-D gridded representation of the STARS data. This is obtained by defining a geographical domain that covers the tracks of PLs (latitudes: 57°N - 83°N, longitudes: 20°E - 66°E) and setting all grid-cells inside the area of observed PLs (track location plus radius) to 1 and all other grid-cells to 0. Diagnostic index values are calculated for each grid-cell in the same geographical domain. For 8 PLs in STARS the approximate radius is not reported. As we require the empirically observed radius for defining the empirically observed area of observed PLs, we sort out these 8 entries to ensure a consistent empirical dataset with 132 reported PLs.

To investigate if the diagnostic indices may be used to distinguish times and locations at risk for PL occurrence, it is important to not only analyse index values at times and locations when PLs have occurred, but also when no PLs have occurred. For

| Name | Definition | Unit | Required dimension of input data |
|---|---|---|---|
| Conventional MCAO index | $m_\theta = \theta_{\text{skin}} - \theta_{850\text{hPa}}$ | K | 2-Dimensional |
| 3-D variant of the conventional MCAO index | $m_\theta^p = \theta_{\text{skin}} - \theta_p$ | K | 3-Dimensional |
| Region-specific variant of the conventional MCAO index | $m_\theta^{\text{crit}} = \theta_{\text{skin}} - \theta_{p_{\text{crit}}}$ | K | 2-Dimensional |
| New MCAO index | $m_p = p_0 - p^*$ | hPa | 3-Dimensional |
| New Polar Low index | $m_{tr} = p^* - p_{tr}$ | hPa | 3-Dimensional |

**Table 1. Diagnostic indices for measuring MCAOs.** The term "conventional MCAO index" summarizes previously used diagnostics that are based on the difference in potential temperature between the ocean surface and a fixed characteristic pressure level aloft, as, e.g., 850 hPa (Papritz et al., 2015; Kolstad, 2017; Polkova et al., 2021), or other levels (Kolstad et al., 2009; Fletcher et al., 2016; Papritz and Sodemann, 2018; Landgren et al., 2019). Notation: $\theta_{\text{skin}}$, $\theta_{850\text{hPa}}$, $\theta_p$, and $\theta_{p_{\text{crit}}}$ denote the potential air temperature at the surface, at 850 hPa, at pressure level, $p$, and at a critical characteristic pressure level, $p_{\text{crit}}$ (see Sec. 3.3.4), respectively; $p_0$, $p^*$, and $p_{tr}$ denote the air pressure at the surface, at the upper boundary of MCAOs and at the dynamical tropopause, respectively. Details of the indices are given in the text (see Eq. 1, 2, and 7 for the conventional MCAO index, its 3-D variant and a region-specific variant obtained by fitting to data about observed PLs; see Eq. 3 and Eq. 4, for the new MCAO and Polar Low indices, respectively).

that purpose, we calculate all diagnostics during a set of randomly selected "pseudo-events". Pseudo-events are defined as hypothetical PL events with the same frequency, and the same temporal and spatial scale as the actual PL events observed in STARS. This provides an experimental setup for testing if diagnostic indices are able to distinguish actual from hypothetical PL events. The pseudo-events are defined as follows: (i) we compute the average duration, track length and radius of all PLs reported in STARS; (ii) we randomly select times during October-March of all years 2002-2011 when no PLs occurred (according to STARS); (iii) we randomly define the genesis location of a pseudo-event somewhere in the geographical domain of interest over open waters; (iv) we define the track of the pseudo-event by randomly defining the main direction and a series of randomized track increments such that the overall track length of all events matches the observed mean track length but individual events exhibit small variations around this (to mimick some of the observed variations in PL tracks); (v) we use the average radius of observed PLs to define the area of a pseudo-event around the randomly generated track. The diagnostic indices are calculated for all random pseudo-events, which serve here as a control set for investigating the robustness of the relationship between diagnostic indices and observed PLs during "normal" weather conditions (randomly chosen days). Two examples of randomly generated pseudo-events are illustrated in Appendix B, Fig. B1.

The performance of diagnostic indices in distinguishing the time and location of occurrence of PLs is assessed by (i) visual analysis and comparison of index maps with maps of empirically observed tracks of PLs; (ii) automated counting of the number of matches between areas with high index values and locations of past PLs; and (iii) a Receiver Operating Characteristic (ROC) curve and accuracy scores - both widely used metrics for binary classification problems (Fawcett, 2006; Tharwat, 2021). ROC curves summarize sensitivity and specificity values for different classifiers and the accuracy score is used to measure how good

a model (in this case: diagnostic indices) predicts the desired classes (in this case: PL or non-PL event).

For comparison of diagnostic indices with observed PLs, we compute for each grid-cell a temporal average, $M_i$, of the hourly index values ($m_\theta$, $m_p$, $m_{tr}$ - as defined in Eqs. 1, 3 and 4 and in what follows also denoted as $m_i$, with $i \in \{\theta, p, tr\}$). The temporal average is computed over a time-interval that covers the reported time-interval of each PL in STARS. For the conventional and the new MCAO index, we compute the simple temporal average $M_i = \frac{1}{T} \sum_{t_0}^{t_{\text{end}}} m_i(t)$, where $t_0 = t^*_{\text{start}} - 12\text{h}$ and $t_{\text{end}} = t^*_{\text{stop}} + 12\text{h}$ are the start and end times of the time-interval around each PL ($t^*_{\text{start}}$, $t^*_{\text{stop}}$ are the reported start and stop dates in STARS). We compute the index values for $\pm$ 12 hours to capture also environmental conditions around the genesis and lysis of PLs (comparable approach regarding the genesis time-interval also used in Terpstra et al. (2016)). For the new PL index we compute the temporal average, $M_{tr}$, by considering only the hourly index values, $m_{tr}$, for hours with lower-level instability ($m_p > 0$). The different temporal average for the PL index, as compared with the MCAO indices, is chosen because a lower-level instability (as indicated by a non-zero MCAO index) is assumed to be a necessary condition for the occurrence of PLs (following Kolstad, 2011) and we wanted to test if additional information about the upper-level anomaly in the areas of MCAOs, as captured in $m_{tr}$, improves the performance of the index compared with the simpler new MCAO index.

## 3   Results and discussion

### 3.1   Interactive 3-D visual analyses of MCAOs and PLs in ERA5

In this section we summarize selected examples from the interactive 3-D visual data analyses. The aim is twofold: (i) illustrate 3-D perspectives on MCAOs and PLs, as represented in ERA5, for contributing to the understanding of these phenomena (Fig. 1, Fig. 2), and (ii) show examples of the application of novel methods for interactive visual exploration of complex meteorological data (Movies 1,2). The high resolution and consistency of ERA5 allows for both, detailed 3-D visual analyses of single cases of MCAOs and PLs (as reported in this section), as well as long-term assessments of diagnostic indices used to capture these events (see subsequent sections). The methods for 3-D IVA that we show here can be applied generically for analyses and visualization of a variety of meteorological phenomena in ERA5 and other gridded meteorological data.

We analyse in 3-D the transport of cold air from regions covered by sea-ice toward the ice-free ocean during MCAOs in ERA5 by assessing, for example, the dynamics of the 3-D isosurface of potential air temperature, illustrated here for two MCAO events in the Barents and the Nordic Seas (Fig. 1, Movie 1). During the MCAO event in December 2002 (Fig. 1c-e), the volume of cold-air grows in vertical extent before moving southward over the sea-ice into the Barents Sea. Complex spatio-temporal structures with great variation in shape and dynamics between different MCAO cases can be observed (e.g. Fig. 1d-f). Comparison with conceptual models and previous examples of standard 2-D depictions of MCAOs (e.g., Gryschka, 2018) underlines that advanced methods for 3-D visual analysis provide more comprehensive insights and can be used for quick and flexible, interactive exploration of data volumes (Movies 1,2). While the visual inspection of cloud cover does

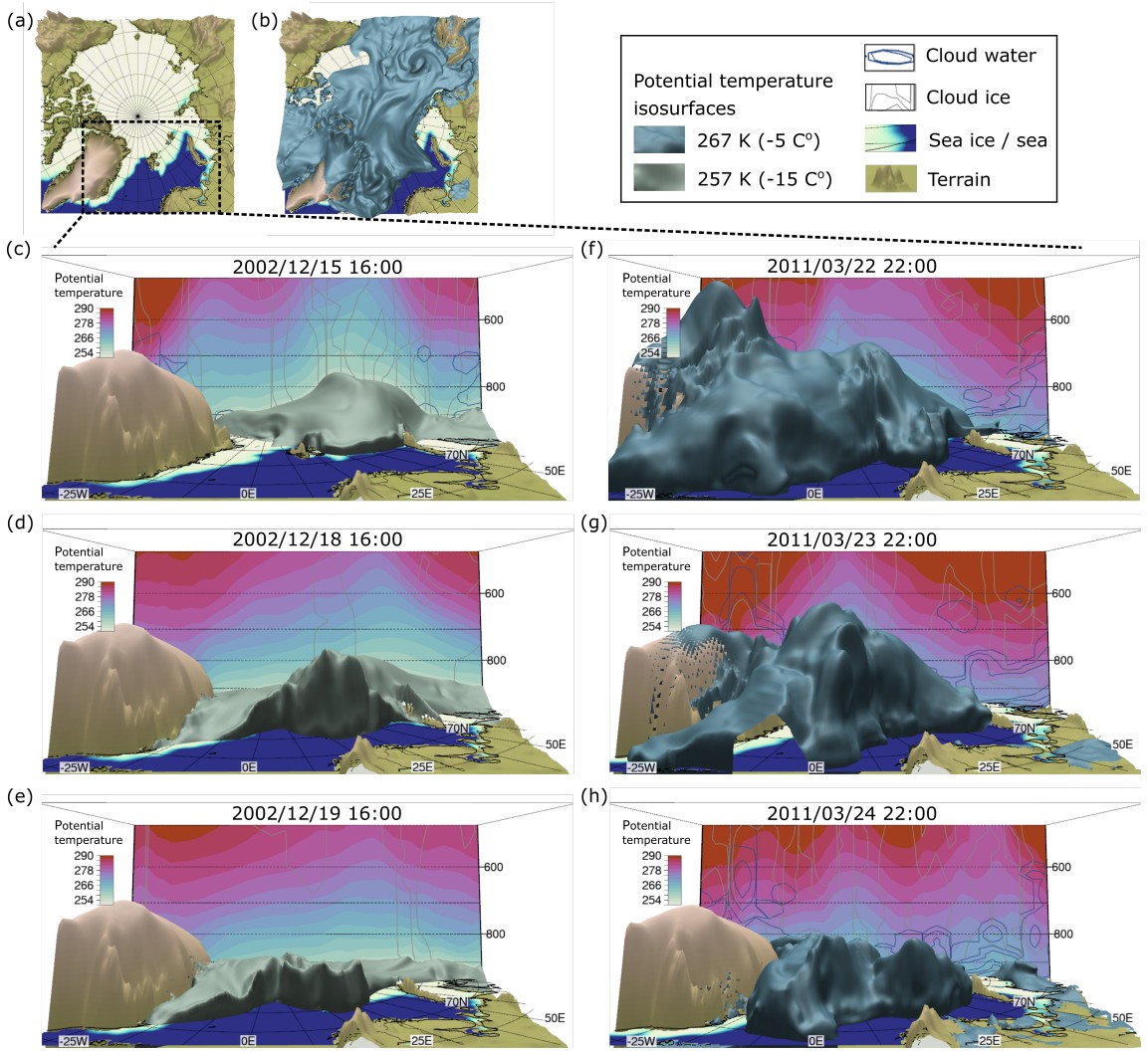

**Figure 1. Three-dimensional interactive visual analysis of MCAOs in ERA5 data.** Cold air is delineated by isosurfaces of potential temperature. The air below the isosurfaces has lower potential temperatures and the air above has higher potential temperatures. **(a - b)** show a top-view for locating the geographical domain of interest (Barents and Nordic Seas). Temporal snapshots illustrate the transport of cold air during two MCAOs: **(c - e)** Case 1, 2002/12/15-19 (257 K isentrope); **(f - h)** Case 2, 2011/03/22-24 (267 K isentrope). Contour lines of cloud water and cloud ice are drawn at $[10^{-5}, 10^{-6}, 10^{-8}]$ kg/kg. Movie 1 demonstrates the interactive 3-D data exploration of MCAO case 1 using *Met.3D*. Fig. 2 illustrates winds during the PLs that formed within the MCAO depicted in panels **c - e**.

partly resemble typical thick cloud bands at the boundary of the outbreak, as described, e.g. in Gryschka (2018), the details of the characteristic 3-D structure of convective cloud bands are not resolved in ERA5 for the cases that we inspected (not shown).

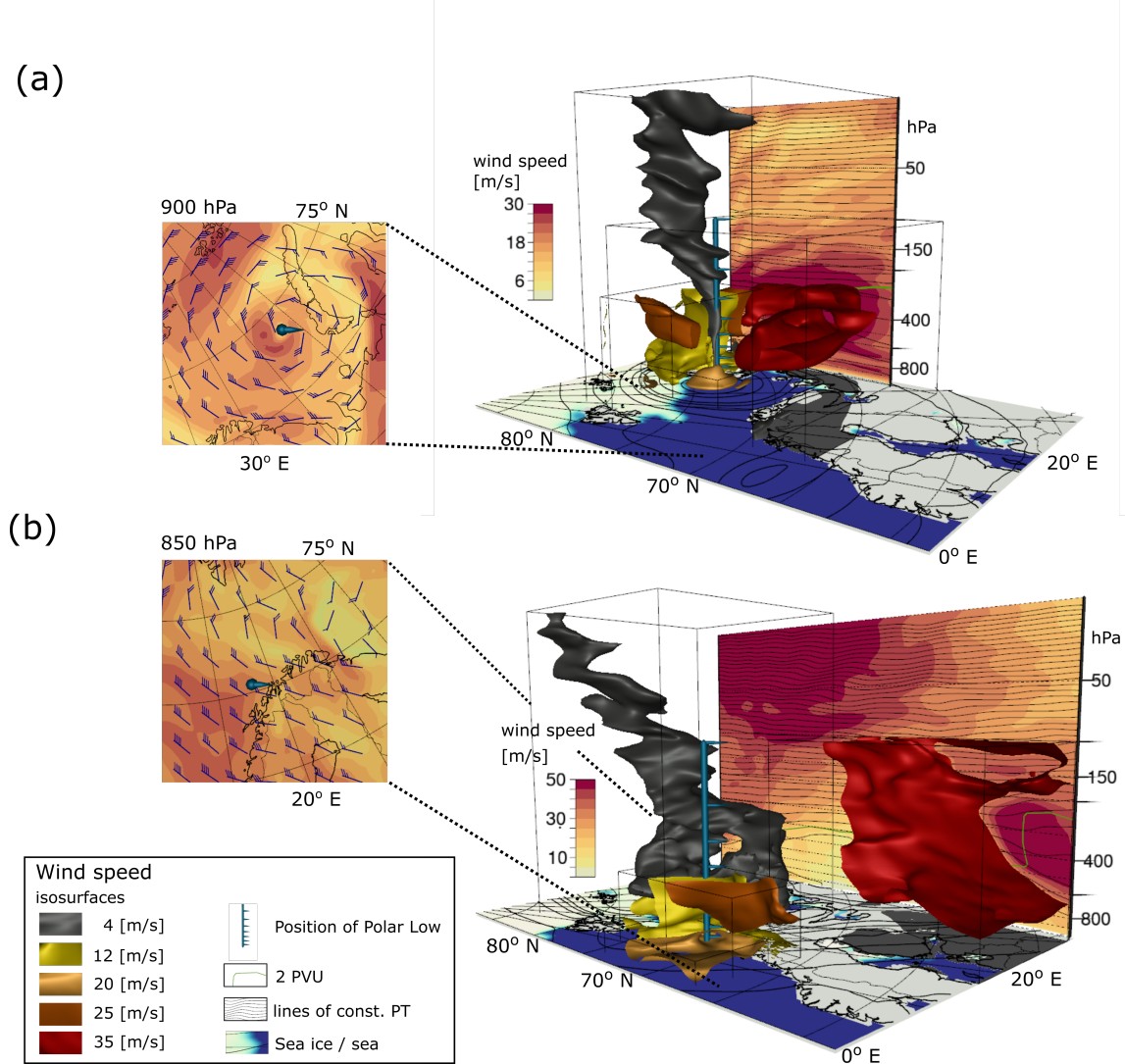

**Figure 2. Three-dimensional interactive visual analysis of PLs in ERA5 data.** Winds are visualized for the times of two PLs. **(a)** In case 1, 2002/12/19, 16:00, a symmetric slow wind "eye" is observed (gray isosurface) in the area where a PL is reported (Bracegirdle, 2012; Føre et al., 2012), with a pronounced vortex in the wind field around it (see inlay). The tube-like, coherent volume of air with slow winds extends up into the stratosphere. As expected, fast winds are observed near the surface. Complex 3-D structures of the wind field in the lower troposphere are visualized by means of isosurfaces constrained to different bounding boxes for selective illustration of various wind speeds in different volumes of air in close proximity to the observed PL. The illustrated aspects of the 3-D wind field show similarities to previously described, average wind-patterns during reverse shear PLs (Fig. 8 in Michel et al., 2018; Terpstra et al., 2016). Movie 2 demonstrates the interactive 3-D data exploration of PL case 1 using *Met.3D*. **(b)** In case 2, 2011/03/24, 11:00, no symmetric slow wind eye is observed in the lower troposphere in the area of the reported PL (Noer et al., 2011). The jet-stream is stronger and located further south.

A PL that was described as an "Arctic hurricane" (Bracegirdle, 2012) formed west of the coast of Novaya Zemlya, during the 19-20th of December 2002, towards the end of the MCAO illustrated in Fig. 1. The capacity for quickly analysing various data variables from different angles using different visualization methods is a key advantage of 3-D IVA, because it makes it easier to explore new perspectives and discover potentially interesting features. One such example emerging from our case-studies is a "slow wind perspective" on PLs (Fig. 2, Movie 2). During some of the PL events we analysed, there is a coherent tube-like volume of air with slow wind speeds at the center of the PL which extends from the surface up into the stratosphere (e.g. winds < 4 m/s in Fig. 2a). This coherent 3-D feature of the wind field is consistent with the 2-D vertical profile of winds during reverse shear PLs (Fig. 8, Michel et al., 2018; Terpstra et al., 2016), but here it is illustrated in 3-D and for a much larger vertical extent. Inspection of isosurfaces at different wind-speeds reveal complex wind flows during PLs, with strong near-surface winds (Fig. 2, Movie 2). Visual analysis of wind-barbs show vortex-like wind fields around the slow-wind eye. Interactive sliding of cross-sections through different vertical levels helps locating altitudes with strong winds as well as regions with strong wind shear (Movie 2). Resolved dynamics of cloud cover during the PL on the 19-20th of December 2002 hint at the observed symmetrical vortex of clouds during this PL (Bracegirdle, 2012), but details are not resolved in ERA5 (not shown).

## 3.2 Interactive 3-D visual analysis of diagnostic indices and vertical structure of MCAOs and PLs

Inspecting the 3-D variant of conventional MCAO indicators ($m_\theta^p$, Eq. 2) unravels the sensitivity to changes in the characteristic pressure level (Sect. 3.2.1). Aiming at avoiding this sensitivity by conceptualizing new diagnostics, we analysed the vertical extent of MCAOs (Sect. 3.2.2) and the vertical distance to the dynamical tropopause (Sect. 3.2.3). Results described in this section served as motivation for the definition of new diagnostic indices (as summarized in Sect. 2.3).

### 3.2.1 Sensitivity of conventional MCAO indices to the choice of the characteristic pressure level

Visual analyses of changes in $m_\theta^p$ (Eq. 2), the 3-D variant of conventional MCAO indices, at different locations when varying the characteristic pressure level show that the magnitude of MCAO index values as well as the geographical area with positive MCAO index values decrease with increasing altitude (Fig. 3, Movie 3). At low altitudes, close to the ocean surface, potential air temperature is lower than potential skin temperature, leading to positive MCAO indices. At higher altitudes, potential air temperature increases, leading to negative MCAO indices (set to zero in Fig. 3a-c, Movie 3). Note that areas with extreme values of the conventional MCAO index, i.e. maximum temperature difference between the sea surface and a certain pressure level, do not necessarily coincide with areas where the conventional MCAO index is positive also at high altitudes (i.e. $m_\theta^p > 0$ at high vertical levels $p$; see Fig. 3, Movie 3). The MCAO index values change substantially depending on the choice of the characteristic pressure level, which suggests reconsidering the thresholds for distinguishing between weak, moderate and extreme MCAOs depending on the choice of the pressure level.

The sensitivity of conventional MCAO indices to the choice of the characteristic pressure level can be expected, considering standard vertical profiles of potential temperature along with a dynamic 3-D shape of the volume of cold air and its complex

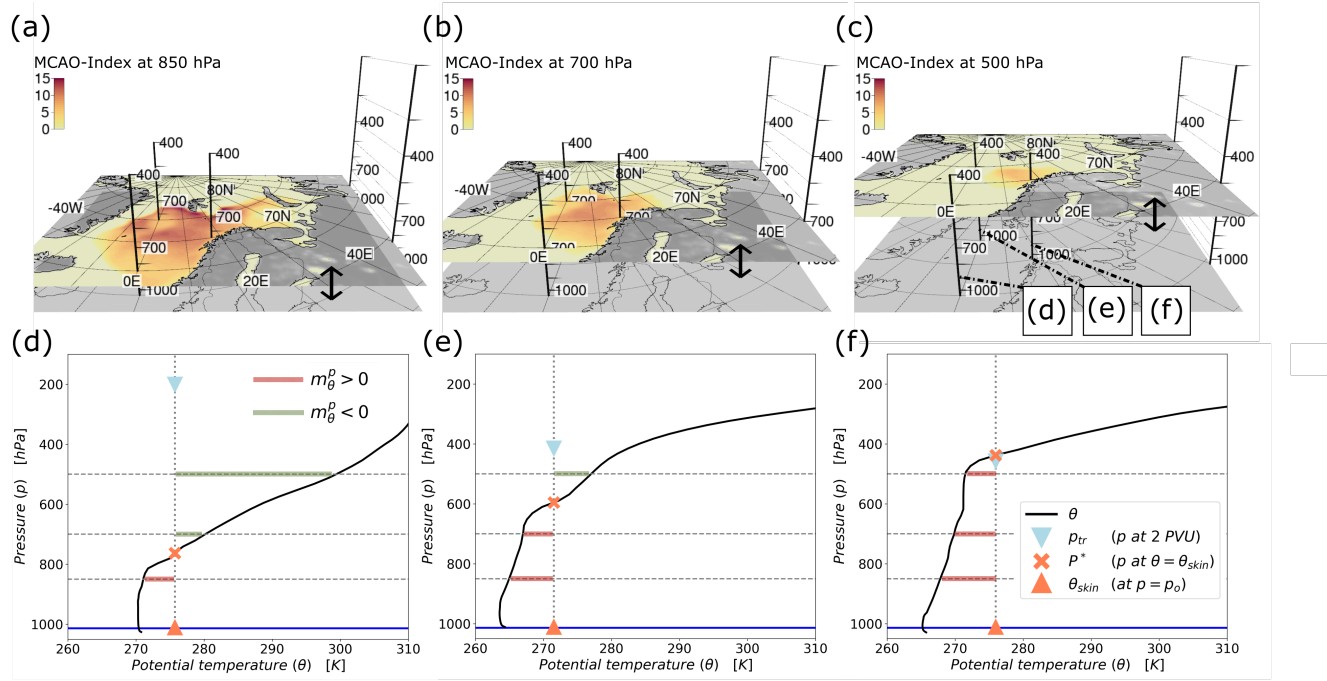

**Figure 3. Extending the conventional MCAO index to 3-D. (a - c)** Sensitivity of the conventional MCAO index to the choice of the characteristic pressure level. The MCAO index is calculated for the 24th of March 2011, 14:00, at different characteristic pressure levels (i.e. Eq. 2 with different values of $\theta_p$): **(a)** 850 hPa; **(b)** 700 hPa; **(c)** 500 hPa. **(d - f)** Vertical profiles of potential air temperature at selected locations within the MCAO (locations indicated by the poles in panels **a - c**). The 3-D variant of the conventional MCAO index is positive below and negative above the critical pressure level, $p^*$ (illustrated as an orange cross), at which $\theta_{\text{skin}} = \theta_p$. The red and green lines illustrate the difference in potential temperature at the pressure levels shown in **a - c**. Movie 3 demonstrates the interactive visual analysis of the effect of varying the characteristic pressure level in the conventional MCAO index. Fig. 4 illustrates the 3-D structure of the MCAO shown here by visualizing the isosurface of $p^*$ for the entire geographical domain.

mixing with air masses above the ocean. With the aim of formulating a diagnostic index that is not sensitive to the choice of a characteristic pressure level, we consider in more detail the 3-D structure of potential air temperature.

### 3.2.2 The upper boundary of MCAOs

The interactive visual analyses of the 3-D MCAO index, along with standard vertical profiles of potential air temperature,
suggests that the vertical profile of potential air temperature in the column of air above each location inside of a MCAO may be sketched as follows: in the lower troposphere, there is an unstable layer of air, in which the potential air temperature decreases with altitude, followed by a layer with approximately constant potential air temperature, and then a stable layer of air in which potential temperature increases with altitude. Fig. 3d-f shows the vertical profile of potential temperature at exemplar locations. This implies that, in the column of air above each location (grid-cell) within the area of a MCAO, there should be at least one

pressure level at which the 3-D MCAO index, $m_\theta^p$, changes its sign. The critical pressure level, $p^*$, at which $m_\theta^p$ changes its sign is the altitude at which potential temperature aloft equals potential skin temperature,

$$\theta_{p^*} = \theta_{\text{skin}}. \tag{5}$$

The column of air below $p^*$ is unstable and the air column above it is stable - with respect to the simple static stability criterion based on the difference in potential temperature aloft and at the surface, as used in conventional MCAO indices (Landgren et al., 2019; Kolstad et al., 2009; Fletcher et al., 2016; Papritz et al., 2015; Kolstad, 2017; Papritz and Sodemann, 2018; Polkova et al., 2019). We therefore define the critical pressure level, $p^*$, as a simple measure of the upper boundary of MCAOs.

The upper boundary of the lower-level instability caused by MCAOs can be visualized by computing the zero-isosurface of $m_\theta^p$. Visual analyses of the dynamics of the zero-isosurface reveal interesting spatio-temporal dynamics in the upper boundary of MCAOs (Fig. 4, Movies 4-6). Investigation of several MCAO cases indicates a trend for the upper boundary of the lower-level instability to increase with distance from the sea-ice edge. This in accordance with conceptual descriptions about strong organized convective processes and convective overturning that cause a vertical increase of the MCAO depth with increasing distance from the sea-ice (e.g., Gryschka, 2018). However, there are substantial spatio-temporal variations during the course of single MCAOs and particularly between different MCAOs. Interestingly, visual comparison of the upper boundary of MCAOs and the position of observed PLs during several of our use-cases shows that geographical areas with the highest vertical extent of MCAOs coincide with geographical areas of PLs.

### 3.2.3 The vertical distance between lower-level instability and upper-level anomaly during PLs

Both a lower-level instability and an upper-level forcing of the dynamical tropopause are required for PL genesis (Kolstad, 2011; Grønås and Kvamstø, 1995). Along these lines, Grønås and Kvamstø (1995) showed that, for 2 out of 4 PLs, the distance between the surface and the top of the atmospheric boundary layer was smaller than 2500 m. In related previous work Kolstad (2011) advocated that there was an association between PLs and the co-occurrence of two factors: a positive MCAO index and an upper-level potential vorticity anomaly. Kolstad (2011) used these factors to define a binary PL index. Motivated by these studies, we visually explore in Met.3D the spatio-temporal variations of the upper boundary of MCAOs ($p^*$, Eq. 5) in combination with spatio-temporal variations of the dynamical tropopause, defined here at 2 Potential Vorticity Units (PVU = $10^{-6}$ Km$^2$kg$^{-1}$s$^{-1}$; see Fig. 4, Movie 5).

Interactive 3-D visual analysis of single cases indicates that the distance between the dynamical tropopause and the upper boundary of MCAOs (Sect. 3.2.2) is smaller in geographic proximity to areas where PLs occur. During some of the PL cases, the dynamical tropopause extends downward into the lower-level instability region induced by MCAOs (Fig. 4e, crossing the zero-isosurface of $m_\theta^p$ with the dynamical tropopause; Movie 5).

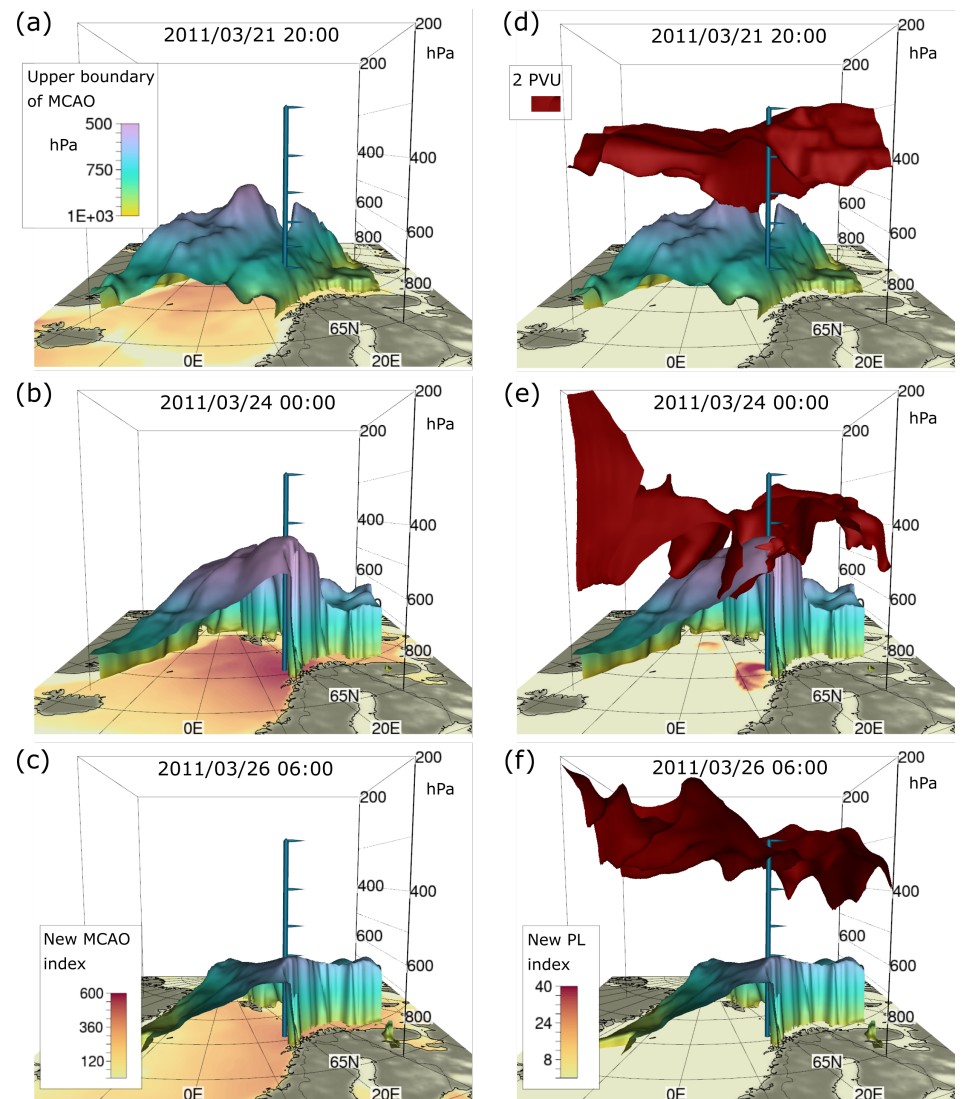

**Figure 4. Upper boundary of a MCAO and its distance from the dynamical tropopause (a - c)** Spatio-temporal variations in the upper boundary of a MCAO, defined here as the pressure level, $p^*$, at which $\theta = \theta_{\text{skin}}$. The new MCAO index, $m_p$ (Eq. 3), which approximates the "height" of MCAOs is shown on the horizontal cross-section. **(d - f)** The dynamical tropopause extends downwards into the lower-level instability caused by the MCAO. Movies 4-6 show the 3-D dynamics of the upper boundary and the dynamical tropopause for this case and for the MCAO case depicted in Fig. 1. The vertical pole is plotted at a fixed location to indicate the mean position of the observed PL in STARS.

### 3.3 Evaluating diagnostic indices by comparison with observed PLs

In previous sections, we summarized our case-studies of MCAOs and PLs along with the definition and motivation for new diagnostic indices. Further on, we evaluate these new indices, along with the conventional MCAO index, by comparison with all PL events observed during years 2002-2011 in the Barents and Nordic Seas (as reported in Noer et al., 2011; STARS - data, 2013). We investigate the distribution of index values in different geographical domains during all PLs (Sect. 3.3.1), the number of matches between areas with high index values and areas, where PLs have been observed (Sect. 3.3.2), and the performance of indices in distinguishing times and locations of observed PLs from times and locations without PLs (Sect. 3.3.3).

#### 3.3.1 Distribution of diagnostic index values in areas within and outside of observed PLs

Analysing the distribution of index values in all grid-cells during all years 2002-2011, within and outside of the areas of observed PLs (defined via the track and radius of PLs reported in STARS), shows that (i) high values of the conventional MCAO index occur only slightly more often in areas, where PLs were observed compared with all other areas in the domain (Fig. 5b); (ii) the new indices differ substantially between the two geographical domains - within and outside of observed PLs (Fig. 5e,h); (iii) high values of the new MCAO index and low values of the new PL index occur more often in geographical areas of the observed PLs compared with the rest of the domain (Fig. 5e,h).

The mean vertical extent of MCAOs within the area of all observed PLs during years 2002-2011, as captured by $m_p$ (Eq. 3), is 289 hPa. Substantially higher vertical extents were observed (Fig. 5e), with a maximum at 607 hPa. In comparison, a mean vertical extent of 161 hPa is observed in areas outside of PLs. The mean vertical extent of MCAOs (289 hPa) corresponds to an upper boundary of the MCAO at $1013.25 - 289 \approx 724$ hPa and the maximum vertical extent corresponds to an upper boundary of the MCAO at approximately 406 hPa.

The mean distance between the upper boundary of the lower-level instability and the dynamical tropopause, as captured by $m_{tr}$, during all PL events is approximately 345 hPa, but substantially smaller distances are also observed (Fig. 5h). Interestingly, in 41 % of all PLs, there is a short time of at least one hour, during which the dynamical tropopause extends downward into the lower-level instability ($p_{tr}(t) > p^*(t)$) within the area of observed PLs (separate analysis not shown in the Figure). In comparison, a larger mean distance of approximately 510 hPa is observed in the areas outside of observed PLs (Fig. 5).

Investigating diagnostic indices during "pseudo-events", defined as randomly selected times when no PL occurred with an area that matches in scale the area of actual PL events (see Sect. 2.4), shows that low values for the new MCAO index ("shallow" MCAOs) and larger values of the new PL index (low or absent forcing from the dynamical tropopause) occur more often during normal weather conditions than during PLs (gray bars in Fig. 5e-f and h-i). This suggests that the new indices capture features that are useful for distinguishing meteorological conditions during PLs from meteorological conditions during a randomly selected set of days in the Nordic winter.

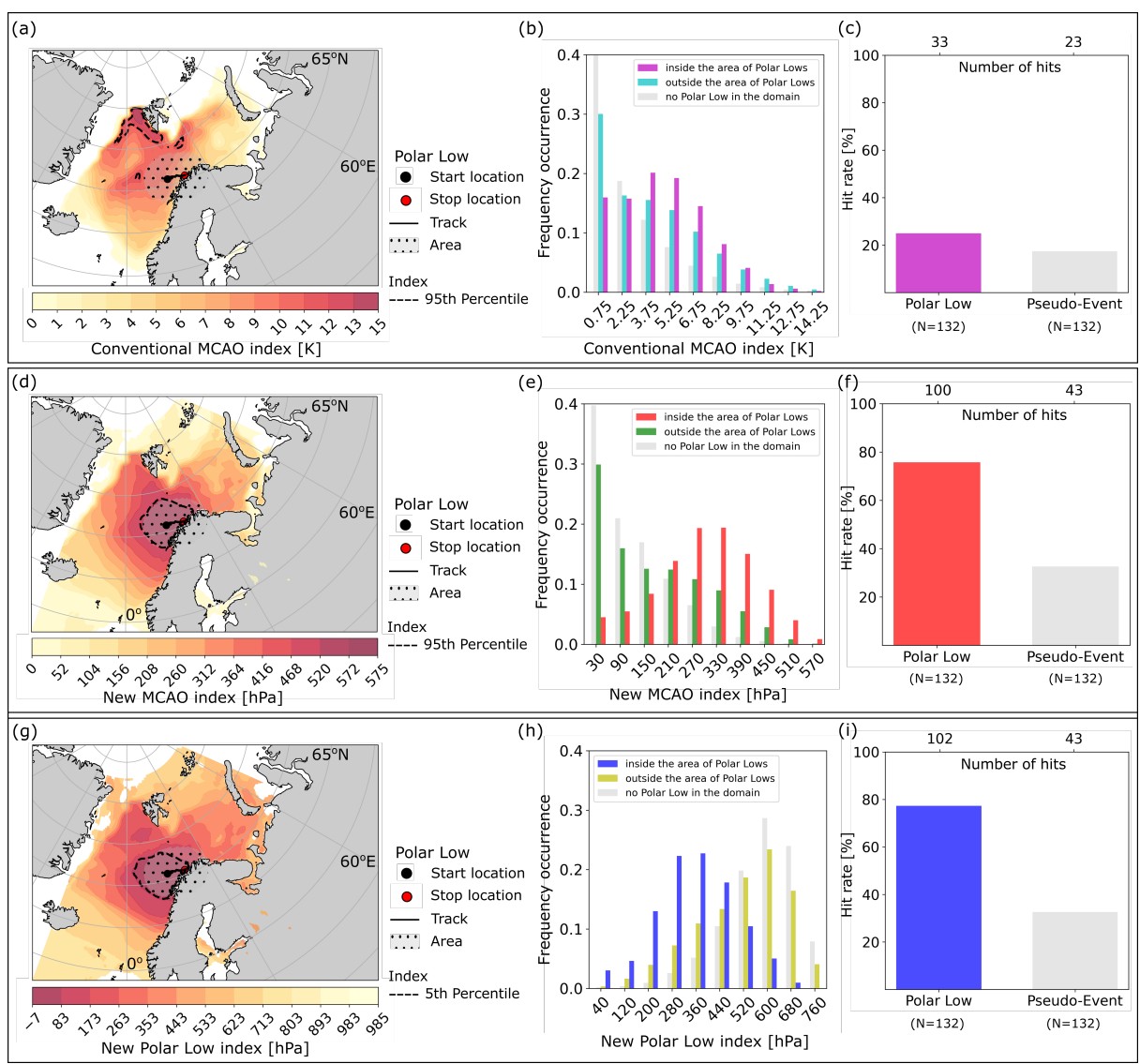

**Figure 5. Association of diagnostic indices with locations of observed PLs. (a - c)** Conventional MCAO index (Eq. 1); **(d - f)** new MCAO index (Eq. 3); **(g - i)** new PL index (Eq. 4). Maps **(a, d, g)**: Diagnostic indices for one selected example, the PL on the 24th of March, 2011 (index values are averaged over all time-steps of the event; see Sect. 2.4). PL track from the STARS dataset (Noer et al., 2011). Histograms **(b, e, h)**: distribution of index values in all grid-cells for all PLs during years 2002-2011 separated into areas within and outside of the area of observed PLs. Gray bars show index values in all grid-cells of the domain during a set of randomly selected times with no PL (pseudo-events defined as a proxy for "normal" weather conditions without PLs; see Sect. 2.4). Bar-charts **(c, f, i)**: number of overlaps between areas with high index values ($\geq$ 95-th percentile; panel i: low index values, $\leq$ 5-th percentile) and the area of observed PLs and random pseudo-events, respectively.

### 3.3.2 Association between diagnostic indices and locations of observed PLs

The locations with high values of the new indices, $m_p$ and $m_{tr}$, resemble the locations of observed PLs for selected cases. For example, the area with high values of the new MCAO index during the MCAO case that we previously illustrated (Fig. 1, 3, and 4) matches the area of the observed PL rather well (Fig. 5), considering that it is a very simple index computed without using knowledge about the location of the PL or any fitting procedures. However, as can be expected considering the simplicity of the diagnostics and the complexity of PL genesis, the indices also miss various observed PLs (see Appendix C, Fig. C1, for

two such examples).

We assessed for how many of all the PLs during years 2002-2011 in the Barents and Nordic Seas (as reported in the STARS data) there is an overlap between geographical areas with high values of the new MCAO index and geographical areas of observed PLs (Fig. 5). The association is also analysed for the new PL index and the conventional MCAO index. We compute

one time-averaged diagnostic, $M_i$, for each PL event (averaged over all hours of the event; see Sect. 2.4), and define areas with "high" index values as the 95-th percentile of all index values in the geographical domain. We count the number of events with overlap of at least one grid-cell between areas with high index values and the area of observed PLs. Results for the new MCAO and PL indices show that, for approximately 100 out of the total 132 observed PLs during years 2002-2011 (i.e., in approximately 76% of all cases), the area with high index values overlaps with the area of observed PLs (Fig. 5). This suggests a

notable association between diagnostic indices and locations of PLs. In contrast, for the conventional MCAO index, areas with high index values only overlap with areas of observed PLs in around 33 out of 132 PLs. Clearly, the association between high index values and locations of observed PLs is stronger for the new index values compared with the conventional MCAO index. PLs occurred more often in areas with high vertical extent of the lower-level instability rather than areas with strong instability at low altitudes. This is further supported by a composite analysis computing the long-term average of the conventional and

the new MCAO index for all observed PLs (Fig. 6a-b).

To assess if the number of matches that we obtain is substantially different to the number of matches one would expect by random choice of high index values somewhere in the geographical region of interest, we count the number of overlaps between the areas of randomly selected "pseudo-events" (Sect. 2.4) and the areas of high index values. This shows that the number of

375 matches between high index values and observed PLs is substantially higher than the number of matches between high index values and randomly chosen pseudo-events (Fig. 5), which provides additional supportive evidence for the robustness of the new diagnostics.

The number of matches between areas with high index values and areas of observed PLs is sensitive to the choice of the

380 threshold for delineating "high" index values. We focus on testing the association between areas with particularly high index values ($\geq$95-th percentile), because these are confined to a fairly small area compared with the total geographical domain of the Nordic Seas, which is a useful characteristic for an index that aims at narrowing down the likely location of PLs. Recall that

MCAOs are relatively large-scale phenomena as compared to PLs. Thus as is, the conventional MCAO index is not a useful PL proxy in predictability studies or for marine services as it indicates too large areas where PLs potentially could occur. Results in this section suggest that the new diagnostic indices are useful and informative for distinguishing the location of PLs, given knowledge about the time of occurrence of PLs. However, for the diagnostics to be useful in predictability studies or marine services as PL proxies, it is necessary to demonstrate that they are able not only to identify locations with higher risk for PLs, conditional on knowledge about the time of occurrence, but that they are able to distinguish times and locations of PLs without any prior knowledge from observations. This is tested in the next section.

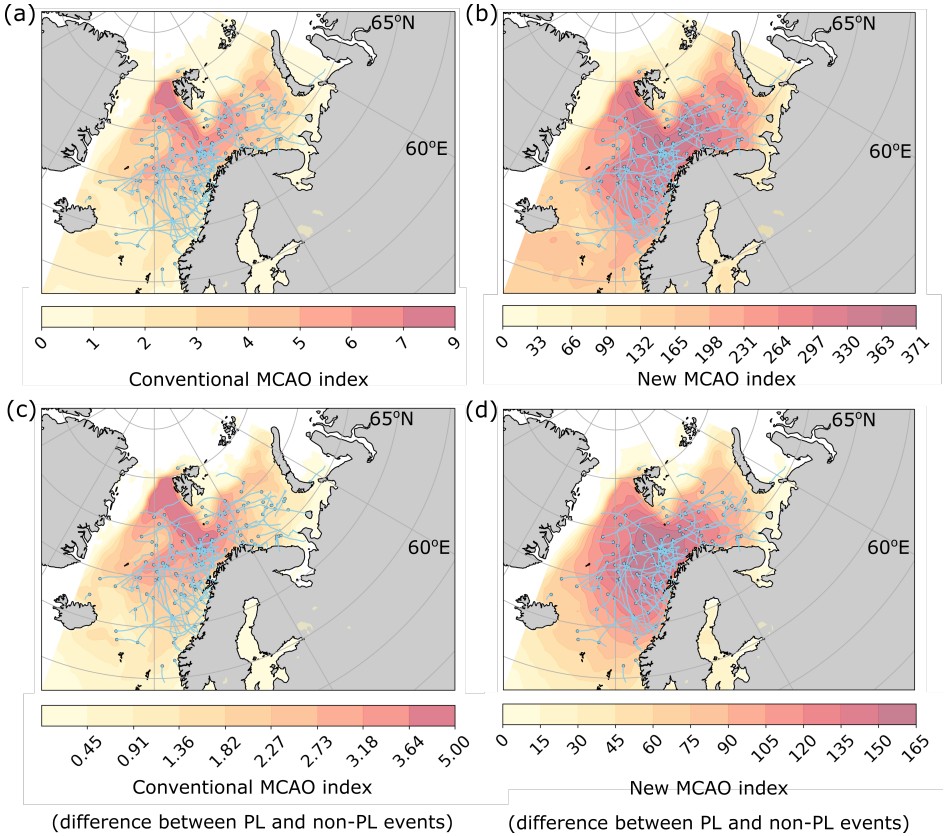

**Figure 6. Composite analysis of the conventional and the new MCAO index. (a)** The conventional MCAO index during all PL events. **(b)** The new MCAO index during all PL events. **(c)** Difference between the conventional MCAO index during PL events and pseudo-events. **(d)** Difference between the new MCAO index during PL events and pseudo-events. PL tracks are plotted as blue lines with the circle at the genesis location reported in STARS.

### 3.3.3 Performance of diagnostic indices in distinguishing the time and location of PLs

For distinguishing times and locations with higher risk for PL occurrence from times and locations with lower risk for PL occurrence based on the simple diagnostic indices, it is necessary that these take on sufficiently different values during PL events compared with no-PL events. As a first step to test this, we conduct a composite analysis for analysing the difference between index values during PL events and non-PL events (random pseudo-events) and compare it visually with observed PL tracks (Fig. 6c-d). The difference between the new MCAO index during PL events compared with non-PL events is particularly large in those areas where many PLs have been observed, whereas the difference is not as large in those areas for the conventional MCAO index (Fig. 6c-d, compare e.g. the area between Iceland and Norway in panels c, d). This indicates that the new MCAO index may serve as an approximate classifier to distinguish times and locations of PLs, whereas the conventional MCAO index is not well suited for this task.

For systematically testing the performance of index values in distinguishing PL occurrence in the observational STARS data, we express the task of distinguishing the time and location of observed PLs as a binary classification problem and measure the skill of index values by means of ROC curves and accuracy scores (details below). For this purpose we introduce for each of the different indices, $M_i$ (with $i \in \{\theta, p, tr\}$, see Sect. 2.4) a set of critical threshold values, $M_i^{\mathrm{crit}}$. The thresholds are chosen to cover evenly distributed the entire range of index values with a constant step-size. For example, $M_p^{\mathrm{crit}} = \{0, 30, 60, ..., 600\}$, to cover the range of values of the new MCAO index (see Fig. 6e) with a constant step-size of 30. Applying the critical thresholds to the index values, we obtain binary indices,

$$
\hat{M}_i = \begin{cases} 1 & \text{if} \quad M_i > M_i^{\mathrm{crit}} \\ 0 & \text{if} \quad M_i \leq M_i^{\mathrm{crit}}, \end{cases} \tag{6}
$$

for all grid-cells during all events (i.e. for observed PLs and for randomly chosen pseudo-events when no PL was observed). We then test the performance in distinguishing PL occurrence based on these binary indices. In the following paragraphs, we first inspect the performance of indices in distinguishing the time of occurrence of PLs (task 1) and second the performance in distinguishing both the time and location of PLs (task 2).

**Task 1: Distinguish the time of PL occurrence**

For distinguishing the time of occurrence of PLs, we use the following classification (summary in Appendix E, Table E1): (i) if the binary index values are positive ($\hat{M}_i = 1$) anywhere in the geographical domain of interest at the time of the event, we define this as a "prediction" that a PL occurs at this time anywhere in the domain (true positives: a PL occurs during that time; false positives: a PL does not occur during that time); (ii) if the binary index, $\hat{M}_i$, is zero everywhere in the domain, we define this as a "prediction" that no PL occurs (true negatives: no PL occurs during that time; false negative: a PL occurs during that time). By computing sensitivity and specificity values for the set of critical threshold values, we obtain one ROC curve for each index (see Fig. 7). The ratio of times that the new index correctly identifies PL events defines the true positive rate (sensitivity) and the ratio of times that the new index incorrectly identifies the PL events as the false positive rate (1 - specificity). For the ROC

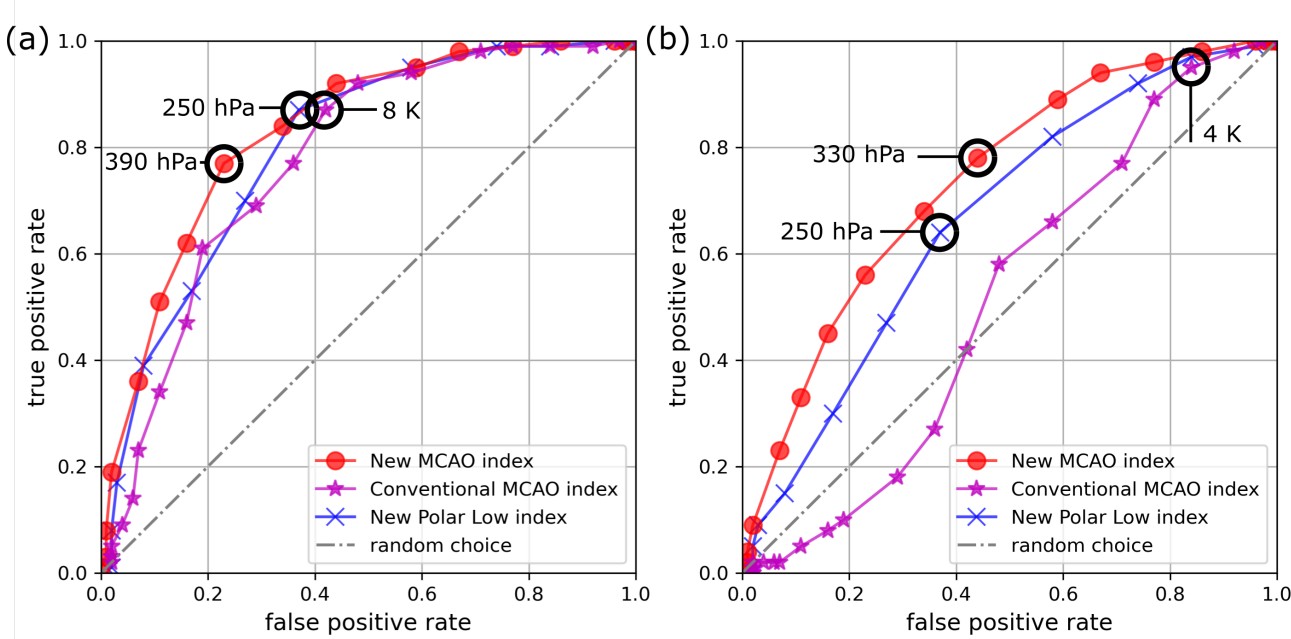

**Figure 7. ROC curves for distinguishing times and locations of PLs based on conventional and new diagnostic indices.** Panels: **(a)** task 1: distinguish the time of PL events and **(b)** task 2: distinguish the time and location of PL events. Index scores are compared with the times and locations of observed PLs, as reported in the STARS data (Noer et al., 2011; STARS - data, 2013), by testing a set of critical thresholds (Eq. 6), for distinguishing areas and times at risk for Polar Lows. Diagnostic indices: the conventional MCAO index (Eq. 1), the new MCAO index (Eq. 3), and the new Polar Low index (Eq. 4). The critical thresholds highlighted in black are the best classifiers, defined as the thresholds $M_i^{\text{crit}}$ (Eq. 6) that maximize Youden's index (Youden, 1950). These are used for calculating the accuracy of indices in distinguishing times and areas of PLs (see text and Table 2).

analysis, the two rates are then plotted against each other (see e.g., Fawcett, 2006). A perfect true positive rate of 1.0 would imply that all PL events are identified correctly. In our experimental setup with 50% observed PL events and 50% non-events, it is necessary that the new index has an area under the ROC curve (AUC score) that is higher than 0.5 for performing better than random chance and being counted as a potentially useful proxy for PLs. All three diagnostic indices perform substantially better than random chance, suggesting that they could be useful for indicating the time of occurrence of PLs (AUC values of 0.78, 0.80 and 0.83, for the conventional MCAO index, the new PL index and the new MCAO index, respectively).

From the set of threshold values, $M_i^{\text{crit}}$, which are used in the ROC analysis for distinguishing the time of PLs, we select the one that maximizes Youden's Index (Youden, 1950), which is defined as sensitivity + specificity - 1, as the best threshold. The best threshold for the conventional MCAO index to distinguish times of occurrence of PLs is $M_\theta^{\text{crit}} = 8$ K. For the new MCAO index, it is $M_p^{\text{crit}} = 390$ hPa, and for the PL index, it is $M_{tr}^{\text{crit}} = 250$ hPa (as shown in Fig. 7a). Using these threshold

values we compute the accuracy score, which is defined as the sum of true positives (TP) and true negatives (TN) divided by all events (see e.g., Tharwat, 2021), for measuring how well the diagnostic indices with the selected binary classifier performs in distinguishing the times of PLs. In our experimental setup, with the same number of PL events as non-PL "pseudo-events", an uninformed random classification would be expected to yield an accuracy score of 0.5. Using the conventional MCAO index with the threshold of 8 K, allows distinguishing the time of occurrence of PLs with an accuracy of 0.73. Using the new MCAO index with the threshold of 390 hPa achieves an accuracy of 0.77. For the new PL index with a threshold of 250 hPa, we obtain an accuracy of 0.75 (all values summarized in Table 2).

**Task 2: Distinguish the time and location of PL occurrence**

For distinguishing the time and location of PLs, we use the following classification (summary in Appendix E, Table E2): (i) true positive, if there is an area with non-zero index values, $\hat{M}_i = 1$ (i.e. $M_i > M_i^{\text{crit}}$), at the time of the event and a PL is observed at that time in that area (overlap of at least 1 grid-cell with the empirical data in STARS); (ii) false positive, if there is an area with positive binary index, $\hat{M}_i = 1$, at the time of the event, but no PL is observed in that area at that time; (iii) true negative, if index values are zero, $\hat{M}_i = 0$, everywhere in the geographical domain at the time of the event and no PL is observed anywhere in the domain at that time; (iv) false negative, if index values are zero everywhere in the geographical domain at a particular time, but there is a PL observed somewhere in the domain at that time.

Results of the ROC analyses show that the conventional MCAO index performs poorly (close to random chance) in distinguishing the time and location of PLs (AUC value of 0.52). This underlines that the magnitude of the conventional MCAO index is not useful for identifying times and locations at risk for PLs. In contrast, the newly introduced indices perform substantially better in distinguishing the time and location of PLs compared with the conventional MCAO index (AUC values of 0.67, and 0.74, for the new PL index and the new MCAO index, respectively). Interestingly, the new MCAO index performs better than the new PL index, despite being simpler and requiring less meteorological input data (the PL index requires the potential vorticity field as an additional 3-D data field). The best threshold value for the new MCAO index that maximizes the sum of sensitivity and specificity in distinguishing the times and locations of PLs in the ROC analysis is $M_p^{\text{crit}} = 330$ hPa (thresholds for the other indices in Fig. 7b). With this threshold, we obtain a sensitivity of 0.78, a specificity of 0.58, and an accuracy of 0.67. In light of the complexity of PL genesis and the simplicity of the new diagnostic index, the aforementioned performance is noteworthy (though still far from maximally attainable reference values of the respective performance metrics). The new MCAO index outperforms the conventional MCAO index and is better than uninformed random choice (all values summarized in Table 2)

In summary, results from testing the performance of the conventional MCAO index and the new MCAO and PL indices (Sect. 3.3.1, 3.3.2, and 3.3.3), suggest that the new MCAO index, $m_p$ (Eq. 3), is a useful alternative to conventional MCAO indices, as it exhibits an association with the times and locations of observed PLs, and hence may serve as a simple proxy for PL

| Index name | task 1: distinguish time of PLs | | | task 2: distinguish time and location of PLs | | |
|---|---|---|---|---|---|---|
| | AUC value | best classifier ($M_i^{\text{crit}}$) | Accuracy | AUC value | best classifier ($M_i^{\text{crit}}$) | Accuracy |
| Conventional MCAO index | 0.73 | 8 K | 0.73 | 0.52 | 4 K | 0.56 |
| New MCAO index | 0.83 | 390 hPa | 0.77 | 0.74 | 330 hPa | 0.67 |
| New PL index | 0.80 | 250 hPa | 0.75 | 0.67 | 250 hPa | 0.64 |

**Table 2. Performance metrics for distinguishing times and locations of PLs based on simple diagnostic indices.** The AUC (area under the curve) values are computed for the ROC curves shown in Fig. 7. The accuracy scores are calculated for the best classifier for each index and task, defined as the classifier that maximizes Youden's index.

occurrence. The new index incorporates more information about the 3-D structure of cold air intrusions from the Arctic and is more skilful in identifying areas and timing favourable for PL development.

### 3.3.4 Determining a region-specific characteristic pressure level from observational data

The new MCAO index was designed to be a simple metric that requires processing of only one 3-D input data field (air temperature) for allowing its use in computationally expensive, long-term assessments that require processing large amounts of data. However, compared with the 2-D conventional MCAO index, the new MCAO index has the disadvantage that it requires more input data. This means that its use in, e.g., predictability studies that compare multiple data sets, might be computationally challenging. In this section, we address this disadvantage of the new MCAO index, along with the disadvantage of the conventional MCAO index regarding the subjective element in the choice of a characteristic pressure level. Both of these challenges can be addressed by determining a characteristic pressure level for the conventional MCAO index from observational data about PLs.

Determining the characteristic pressure level for the conventional MCAO index from observational data results in a region-specific MCAO index that has the same form as the conventional MCAO index,

$$m_\theta^{\text{crit}} = \theta_{\text{skin}} - \theta_{p_{\text{crit}}}, \tag{7}$$

but is based on the *critical* characteristic pressure level, $p_{\text{crit}}$, that maximizes the link to observed PLs. The critical characteristic pressure level can be determined directly from the best classifier ($M_p^{\text{crit}}$) for the new MCAO index,

$$p_{\text{crit}} = p_0 - M_p^{\text{crit}}. \tag{8}$$

The region-specific MCAO index, $m_\theta^{\text{crit}}$, has the advantage that it is computationally cheaper than the new MCAO index, $m_p$, as it can be calculated based only on 2-D meteorological data fields, at the ocean surface and at $p_{\text{crit}}$, while maintaining the same skill in distinguishing the times and locations of PLs as compared to the new MCAO index. Table 1 summarizes key differences between the region-specifc MCAO index and the other diagnostics considered in this study.

The equivalence in skill to distinguish PLs from non-PLs of the region-specific index, $m_\theta^{\text{crit}}$, and the new MCAO index, $m_p$, is evident from the definition of the indices. Similar to the conventional MCAO index, the region specific index has positive values if the potential skin temperature is larger than the potential temperature aloft. The region-specific MCAO index takes on positive values in all grid-cells, in which the new MCAO index, $m_p$, is higher than the threshold value, $M_p^{\text{crit}}$. This is the case because in these grid-cells the critical pressure level, $p_{\text{crit}}$, is vertically located below the upper boundary of the MCAO.

The upper boundary of the MCAO, $p^*$, is defined as the vertical level, at which potential temperature aloft equals potential skin temperature. As potential temperature increases with height in the top vertical layers of the MCAO, the potential temperature at the critical pressure level below the upper boundary of the MCAO is lower than the potential temperature at the upper boundary, and hence also lower than the potential skin temperature, which means that the region-specific MCAO index is positive in these grid-cells (in summary: if $m_p > M_p^{\text{crit}}$, then $p_{\text{crit}} > p^*$, which implies that $\theta_{p_{\text{crit}}} < \theta_{p^*} = \theta_{\text{skin}}$, so that $m_\theta^{\text{crit}} > 0$). This

was confirmed by numerical analysis (see Appendix D, Fig. D1). As the skill for distinguishing PLs from non-PLs is based on the binary index values obtained via the critical thresholds (see Eq. 6), the skill based on $m_p > M_p^{\text{crit}}$ is equivalent to the skill based on $m_\theta^{\text{crit}} > 0$.

The region-specific MCAO index introduced in this section with a threshold of 0 can be used just as well as the new MCAO

index with a threshold of $M_p^{\text{crit}}$ for distinguishing the times and locations of observed PLs. While the procedure of determining the region-specific MCAO index requires processing of the 3-D potential temperature field, this procedure only has to be conducted once in a baseline study for a specific geographical region. The critical characteristic pressure level obtained here is $p_{\text{crit}} = p_0 - M_p^{\text{crit}} = 1013.25 - 330 = 683.25$ hPa for the geographical region of the Barents and the Nordic Seas. For other geographical regions and observational data about PLs, the calculation should be repeated to account for regional differences.

The resulting region-specific MCAO index, with the parameter $p_{\text{crit}}$ "fitted" to empirical data, is computationally cheap and thus feasible for use in climatological assessments and for quick operational risk assessments as part of marine services.

## 4   Conclusions

In the first part of the investigation, we conduct case-studies of MCAOs and PLs in ERA5 data by means of interactive 3-D visual exploration. In the second part, we conceptualize alternative diagnostic indices for MCAOs and PLs, based on insights

from the 3-D IVA. In the third part, we evaluate the performance of the new diagnostics, in comparison with previously used metrics, in distinguishing the times and locations of observed PLs.

We provide a showcase for the potential of 3-D interactive visual data exploration as part of the scientific workflow for advancing the understanding on meteorological phenomena. We reveal complex 3-D shapes of MCAOs and PLs and provide insights

into features as detailed as the slow-wind eye of a PL (Sect. 3.1, Movies 1,2). The interactive analysis with Met.3D underlines the sensitivity of the conventional MCAO index to the choice of the characteristic pressure level (Fig. 3, Movie 3). 3-D IVA improves our understanding of the involved processes and their interaction (e.g., the spatio-temporal dynamics of the upper

boundary of MCAOs and its interplay with the dynamical tropopause, Movies 4-6). Results from the 3-D IVA are used as inspiration for the conceptualization of alternative diagnostic indices (Sect. 3.1). Exploratory visual analyses and a long-term

assessment (2002-2011) of the alternative indices underline that the vertical extent of the lower-level static instability caused by MCAOs, as well as the distance between the lower-level instability and the dynamical tropopause, can serve as improved indicators for PL tracks compared to a widely used 2-D MCAO index (Landgren et al., 2019; Kolstad et al., 2009; Fletcher et al., 2016; Papritz et al., 2015; Kolstad, 2017; Polkova et al., 2019; Papritz and Sodemann, 2018).

We investigate the link between MCAOs and PLs by assessing the performance of MCAO indices in distinguishing times and locations of observed PLs (Sect. 3.3). The quantitative association between MCAO height and PLs shown here could be a first step towards improving marine services. For instance, Polkova et al. (2021) report a prediction skill for MCAOs in the Nordic Sea for as long as 20 days ahead, whereas PLs can only be predicted up to a day ahead. If it were possible to obtain information about e.g. the frequency of PLs from characteristics of MCAOs, this could improve marine services accordingly.

Previous studies demonstrated that MCAOs are a necessary condition for PL development (Ese et al., 1988; Noer et al., 2011; Kolstad, 2011; Mallet et al., 2013). However, what it is exactly about MCAOs that is decisive for PL occurrence and how often PLs accompany MCAOs is still debated. Terpstra et al. (2020) suggested that neither the duration nor the maturity of MCAOs has a distinct relation to PL initiation. Our results might appear at odds with studies suggesting that PLs mostly happen outside of MCAOs (e.g., Terpstra et al., 2016). However, other authors (Rasmussen, 1983; Kolstad, 2011) have pointed out that PLs

happen also within MCAOs, especially those with warm-core thermal instability typically develop deep inside the polar air mass. Upper-level forcing in the form of a positive potential vorticity anomaly is generally associated with the upward doming of underlying isentropic surfaces (see e.g., Hoskins et al., 2007). The height of MCAOs (defined here as the level at which potential temperature aloft equals potential temperature at the surface) therefore depends also on upper-level anomalies, which means the new MCAO index implicitly contains additional information about upper-level forcing that conventional index vari-

ants do not include (as these are purely a measure of the coldness of the CAO relative to the sea surface). This could explain why the association between the new MCAO index and the time and location of PLs is stronger, compared with conventional index variants. The new MCAO index introduced here allows narrowing down the areas at risk for PLs, overcoming one of the disadvantages of the conventional MCAO index, which tends to identify too large areas at risk. Recently, and independently from our study, Terpstra et al. (2021) as well identified the usefulness of considering the vertical extent of MCAOs. The new

MCAO index introduced here closely resembles a metric independently introduced in their study. Both studies underline the usefulness of considering the vertical extent of MCAOs.

Results here suggest that the new MCAO index (Eq. 3) is a promising candidate for use in long-term assessments on seasonal and climatological time-scales as it requires less input data (3-D temperature field and 2-D fields of surface pressure and

555 skin temperature) than the new PL index and shows a slightly better performance (Sect. 3.3). Alternatively, we also suggest a method to determine a critical characteristic pressure level for the conventional MCAO index from empirical data about PL occurrence (Sect. 3.3.4). This results in a region-specific MCAO index that has the same form as the conventional index, but

provides a region-specific quantitative link to observed PLs. It is a good alternative to the new MCAO index, as it is computationally cheaper and thus a more promising candidate for use in seasonal and climatological assessments, which usually compare the phenomena in multiple data sets over very long time-scales. An interesting task for future studies is to compare the predictability of the conventional and the new indices in seasonal prediction systems.

The assessment of the performance of conventional 2-D and new 3-D MCAO indices as proxies for PLs in the Barents and Nordic Seas (STARS data covering the time-period 2002-2011) shows that the new MCAO index performs better in distinguishing time and location of PLs compared with the conventional index (Sect. 3.3). While we show a statistical association between areas with high index values and PLs in a limited geographical and time-domain, our results (e.g. AUC and accuracy scores in Sect. 3.3) also clearly highlight that the complex genesis of PLs cannot be fully captured with the simple diagnostic indices analysed here (both conventional and new indices). This can be expected considering the interplay of various factors that lead to PL genesis. More complex classification schemes, such as self-organising maps, have been also reported recently (Stoll et al., 2021).The new MCAO index introduced here does not capture sea-air sensible heat fluxes, which are captured by the conventional MCAO index (Papritz et al., 2015). We have not distinguished between reverse shear and forward shear PLs in this analysis. Whereas forward shear PLs are the most common ones, the reverse shear PLs are more easily detectable due to strong static instability conditions in the lower troposphere (Michel et al., 2018). Ultimately, the choice of an appropriate diagnostic depends on the objective of the study.

In our analysis, we consider a set of pseudo-events for analysing the behaviour of indices during randomly selected weather conditions. A broader climatological assessment, computing hourly indices for time-intervals of several decades is beyond the scope of this study. Considering that PLs are rare events, the association between areas and times with high index values with areas and times of observed PLs might be overestimated. Future studies could build on this analysis, for example, by analysing longer PL data sets, such as provided in Rojo et al. (2019), taking into account different geographical areas and considering alternative performance metrics, such as the extremal dependence index (EDI, Wulff and Domeisen, 2019). If the performance of diagnostic indices shown in this study for a limited time-interval (2002-2011) and geographical region prove to be robust in other times and regions, then the new indices might be a useful complement for marine services on PLs.

The methods for 3-D interactive visual analysis of ERA5 data introduced here (Sect. 2.2, Sect. 3.1) are publicly available (Met.3D - Homepage, 2021; Met.3D - Documentation, 2021; Met.3D - Documentation ERA5, 2021) and can be used generically, for interactive visual analysis of meteorological phenomena resolved in ERA5 data. We see great potential in using methods for interactive 3-D visual data exploration during the explorative phases of scientific workflows, as performed in this study, for detailed meteorological case-analyses, diagnosis of model simulations and development of new hypotheses.

*Code availability.* The code of the open-source visualization framework *Met.3D* is available at: https://gitlab.com/wxmetvis/met.3d. Links to documentation and further resources are availabe at: https://met3d.wavestoweather.de and https://collaboration.cen.uni-hamburg.de/display/Met3D. The Python and bash scripts for pre- and post-processing, statistical analyses and visualizations are available upon request.

*Video supplement.* The following movies illustrate interactive visual data analysis using *Met.3D* and provide supplementary insights into the
3-D dynamics of MCAOs and PLs in ERA5:

- **Movie 1**: Interactive visual data analysis of a Marine Cold Air Outbreak in ERA5 data.
- **Movie 2**: Interactive visual data analysis of a Polar Low in ERA5 data.
- **Movie 3**: Interactive visual data analysis of the characteristic pressure level in the conventional MCAO index.
– **Movie 4**: Dynamics of the upper boundary of a MCAO (time of MCAO: March, 2011).
- **Movie 5**: Dynamics of the upper boundary of a MCAO and the dynamical tropopause (time of MCAO: March, 2011).
- **Movie 6**: Dynamics of the upper boundary of a MCAO (time of MCAO: December, 2002).

*Author contributions.* Conceptualization: IP, MR, MM, JB; Study design and investigation: MM, IP, MR; Method development (*Met.3D*, Python, bash): MM, KRM, MR; Data Curation and Formal analysis: MM; Validation: MM, LS, IP, MR; Supervision: IP, MR, JB, SO;
Visualization: MM, MR; Writing - original draft and preparation: MM; Writing - review and editing: MM, IP, MR, LS, KRM, JB, SO; Funding Acquisition: SO, JB, MR.

*Competing interests.* The authors declare that they have no conflict of interest.

*Acknowledgements.* We are grateful for the funding by the Deutsche Forschungsgemeinschaft (DFG, German Research Foundation) under Germany's Excellence Strategy – EXC 2037 'CLICCS - Climate, Climatic Change, and Society' – Project Number: 390683824, contribution
to the Center for Earth System Research and Sustainability (CEN) of Universität Hamburg. We acknowledge funding provided by the Blue-Action project from the European Union's Horizon 2020 research and innovation programme under grant agreement No 727852. IP also acknowledges funding from DFG, Project number 436413914. KRM was funded by sub-project Z2b of the DFG Transregional Collaborative Research Centre (SFB/TRR165) "Waves to Weather". We would like to thank the German Climate Computing Center (DKRZ) for providing an excellent research infrastructure, and Frank Toussaint for guidance regarding the ERA5 data archive. Many thanks to Daniel Runow from
the Basis-Infrastruktur Group at the Regional Computing Center, Universität Hamburg, for maintaining the virtual GPU setup that was used for conducting parts of the analyses described here. We highly appreciate the detailed comments from the referees during the review process, which have greatly improved the manuscript.

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

## Appendix A: Summary technical workflow

For the main part of our analyses, we use data from the ERA5 archive hosted at the German Climate Computing Center (DKRZ), which is a copy of the original ERA5 archive (a total of 1.3 petabyte; DKRZ, 2020). The ERA5 data is remapped to a regular latitude-longitude grid using the Climate Data Operators (CDO) (Schulzweida, 2020), as *Met.3D* requires a regularly spaced horizontal grid. The remapping of ERA5 was conducted on *Mistral*, the super-computer at the DKRZ. The remapped data was transferred to the regional computing center for visual data analysis of single cases. For interactive visual analysis in *Met.3D* all data that is rendered into a 3-D scene has to fit into the memory of a single GPU. New data is loaded on demand, when selecting new variables or manually stepping through time. For the visual data analysis, we used a NVIDIA Tesla T4 GPU in a vGPU setup with a global memory of 8 GB. A total of around 3 TB of ERA5 data was probed by means of manual selection and visual data exploration. The remapping of ERA5 data and the computation of the diagnostic indices for the 132 PL events and the 132 random pseudo-events were conducted on *Mistral* at the DKRZ. After selection of the required geographical sub-domain (Barents and Nordic Seas), time-intervals (covering all events) and remapping to regular grids, a total of approximately 1 TB ERA5 data were processed for computing the diagnostic indices.

## Appendix B: Examples of randomly generated pseudo-events

(a)                                                                    (b)

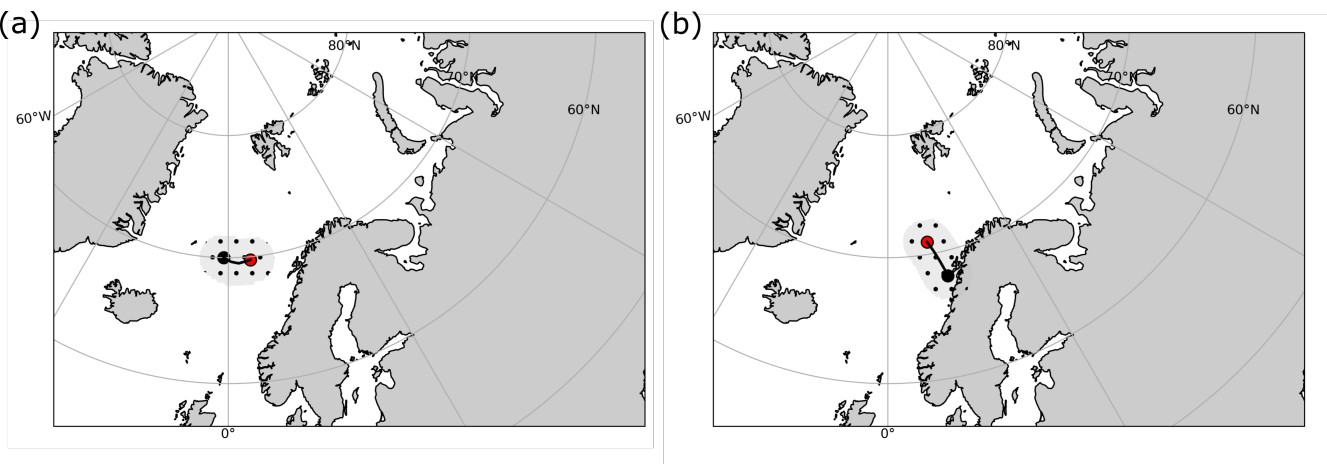

**Figure B1. Examples of randomly generated pseudo-events to test performance of diagnostic indices.** (a) one of the pseudo-events generated in winter 2006 (b) one of the pseudo-events generated in winter 2007. Pseudo-events are defined at randomly selected times and locations when no PL was observed to test if diagnostic indices are useful for distinguishing ocurrence and non-occurence of PLs.

 **Appendix C:  Examples of PL events not captured by the new MCAO index**

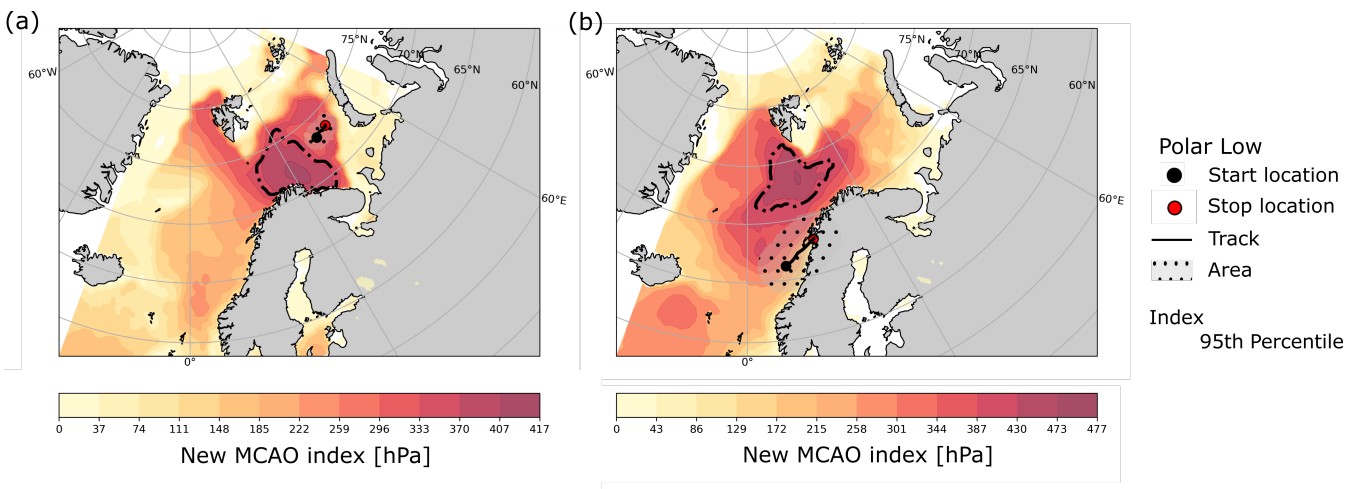

**Figure C1. Examples of PL events that were not captured by the new MCAO index.** (a) PL event on March 4th, 2010 (b) PL event on March 13th, 2011. The area with highest index values, delineated by the 95th percentile (dashed black line), does not overlap with the area of observed PLs (STARS).

**Appendix D: Comparison of dichotomized regional and new MCAO index**

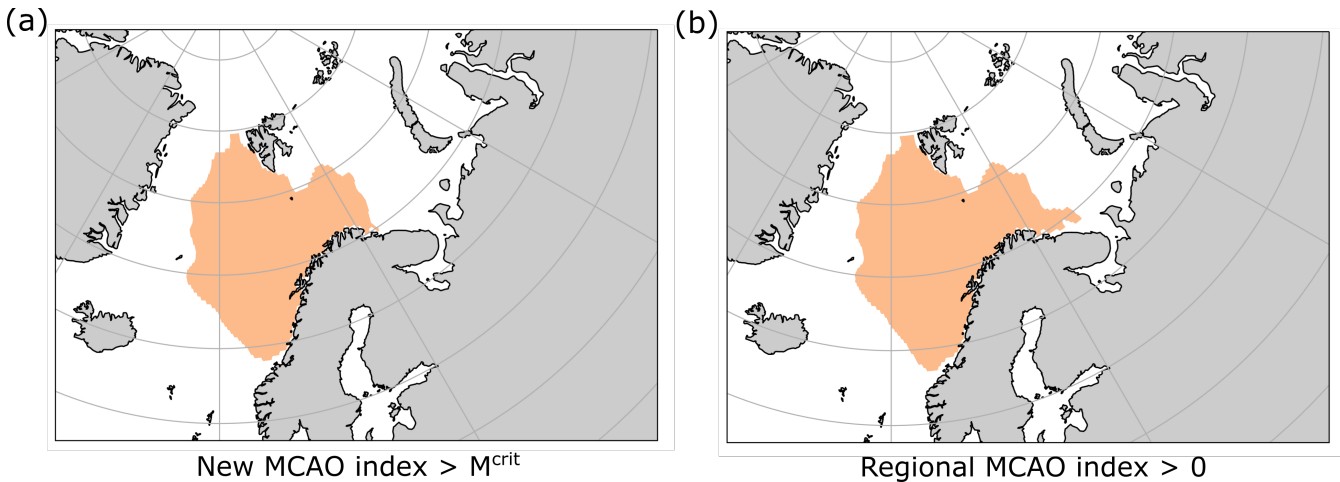

**Figure D1. Comparison of dichotomized regional and new MCAO index for one examplar time-step (14:00 on the 24th of March, 2011).** (a) New MCAO index ($m_p$, Eq. 3). Orange area: the new MCAO index is larger than the critical threshold determined via the analyses in Sect. 3.3.3 (i.e. $m_p > M_p^{\mathrm{crit}} = 330$hPa). (b) Regional index ($m_\theta^{\mathrm{crit}}$, Eq. 7). Orange area: the regional index is larger than zero. As described in Sect. 3.3.4, the critical pressure level for computation of the regional index can be obtained as $p_{\mathrm{crit}} = p_0 - M_p^{\mathrm{crit}} = 1013.25 - 330 = 683.25$ hPa. The comparison provides a numerical example to illustrate that $m_p > M_p^{\mathrm{crit}}$ implies $m_\theta^{\mathrm{crit}} > 0$ (assuming that potential temperature increases with height in the top layers of MCAOs). This means that the skill of the new MCAO index in distinguishing times and locations of PLs with a critical threshold of 330 hPa also holds for the regional MCAO index with a critical threshold of 0, because the classification is based on the dichotomized index values (Eq. 6, Sect. 3.3.3). Note that for the example illustrated here we compute the regional index based on ERA5 data on the nearest pressure level ($700$hPa $\approx 683.25$hPa) to show that this may be used as a computationally cheaper approximation. The new MCAO index is calculated based on ERA5 data on model levels with interpolation to the pressure level where $\theta_{p^*} = \theta_{\mathrm{skin}}$. This explains the small deviations between the two orange areas.

# Appendix E: Classification scheme for testing performance of diagnostic indices

|  | prediction (index values) | |
|---|---|---|
| | PL<br>($\hat{M}_i = 1$, anywhere in domain) | no PL<br>($\hat{M}_i = 0$, everywhere in domain) |
| observation (STARS data)   PL | true positive | false negative |
| observation (STARS data)   no PL | false positive | true negative |

**Table E1. Confusion matrix summarizing the classification scheme for task 1**. In task 1 we test the performance of indices in distinguishing the time of PL occurrence.

|  | prediction (index values) | |
|---|---|---|
| | PL<br>($\hat{M}_i = 1$, inside the area of PL/pseudo-event) | no PL<br>($\hat{M}_i = 0$, everywhere in domain) |
| observation (STARS data)   PL | true positive | false negative |
| observation (STARS data)   no PL | false positive | true negative |

**Table E2. Confusion matrix summarizing the classification scheme for task 2**. In task 2 we test the performance of indices in distinguishing the time and location of PL occurrence.