# Peer review of "Interactive 3-D visual analysis of ERA5 data: improving diagnostic indices for marine cold air outbreaks and polar lows"

_Weather and Climate Dynamics, 2021_

## Referee Comment (RC1)

Review of manuscript 2021-020 submitted to Weather and Climate Dynamics

**Interactive 3-D visual analysis of ERA 5 data: improving diagnostic indices for marine cold air outbreaks**

by Meyer et al.

This study investigates the usefulness of conventional marine cold air outbreak (MCAO) indices, which are typically based on the sea-air potential temperature difference, to identify regions where polar lows (PLs) develop and seeks to improve them in terms of their skill in predicting favourable regions for PL occurrence. For that purpose, the study advocates the use of 3D visualizations of MCAOs and PLs, informing the formulation of alternative indices. Based on case study analyses, two indices are then presented that perform particularly well in terms of predicting PL occurrence as demonstrated for a set of PLs from the STARS database and randomly chosen reference samples. The first index measures the depth of the unstable MCAO air mass by identifying the uppermost convectively unstable level, the MCAO top height, thereby following other studies (e.g., Terpstra et al. 2021). The second index also takes the role of upper-level forcing into account by considering the distance of the MCAO top height and the tropopause, which to my knowledge is novel.

While the the presentation of figures is of high quality and, especially the 3D ones, are simply beautiful to look at, the writing of the manuscript and, in particular, also its organization need improvement. Also I am a bit unsure about novelty and scope of the study, as detailed below. Despite of that I think this study has the potential to become a valuable contribution to WCD if the manuscript is appropriately revised.

**General comments:**

- One of the difficulties I have with this manuscript is whether it fits into the scope of WCD. On one hand, it introduces a workflow for 3D visualization of meteorological fields using the Met.3D software. While I have no doubt that 3D exploration of meteorological phenomena is highly useful and inspiring for formulating hypotheses, introducing such a workflow does not fall into the scope of WCD and other journals might be more appropriate (e.g., Geoscientific Model Development). On the other hand, the specific application, namely identifying better indices for capturing regions favourable for PL development, definitely falls into the scope of WCD. The way it is presented now, the paper tends towards both directions. I strongly suggest the authors focus the paper more towards the meteorological aspectes and less towards a 3D visualization workflow to make it more appropriate for WCD (of course keep the beatiful figures!).

- I also have some reservations regarding the novelty of the presented work. Other studies (see Papritz and Spengler 2017 and Terpstra et al. 2021) have considered the depth of the unstable layer for identifying MCAOs before. In particular, Terpstra et al. (2021) also explore its usefulness for predicting PL occurrence. In addition, while the second index,

which also takes the tropopause level into account, is – to my knowledge – novel, it bears some resemblance to the criterion developed by Kolstad (2011). What are the differences and advantages of your index? Why another index?

- The manuscript is partly not very clearly organized, this applies especially to the methods section, which at great length explains the workflow including many technical and often irrelevant details (such as the supercomputer used) and unnecessary statements à la "We have implemented a functionality in Met.3D…" but omits important information about the methodology. Some unclear points are:
    - Definition of the MCAO indices: these indices should be defined in the methodology section and not in the main part of the paper.
    - Definition of "pseudo-events": How are the spatial regions chosen?
    - For assessing the skill of the MCAO indices it is critical that in the reference sample no PLs occurred. How did you check that during the random time steps there are really no PLs? As I understand the STARS database contains only selected PL cases in the region but by far not all. For example, Michel et al. (2018) find about 243 PLs per winter season, compared to only about 13 PLs per winter in STARS.
    - How are the 95$^{th}$ percentiles computed? Are they spatial or temporal? If they are spatial, what regions are considered?

- The new MCAO index is evidently better suited for identifying regions favourable for PL development than conventional MCAO indices. However, it also has a clear disadvantage: it does no longer take the coldness of the air mass relative to the sea surface into account. Thus, it looses the useful property of conventional indices that they are directly proportional to the sea-air sensible heat flux (cf. Papritz et al. 2015). Hence, it is a matter of application which index is better suited. I would wish the authors include a discussion of this aspect.

**Specific comments:**

- L57: please explain the term potential skin temperature, this may be unfamiliar to non-experts
- L64ff, discussion of different MCAO indices: I miss two important aspects here. First of all, CAOs are convectively unstable air masses subject to strong vertical mixing. This becomes also quite evident in the selected profiels that you show in Fig 4, where the variation of potential temperature with height is small within the CAO air compared to the tropospheric lapse-rate further aloft. Accordingly, the potential temperature is fairly uniform in the vertical, strongly reducing the sensitivity to the choice of characteristic pressure level. Second, there is a clear preference for choosing a lower level because CAO air masses tend to be quite shallow initially and their depth only grows with fetch from the ice edge due to convective overturning and the associated entrainment of air from aloft. Hence, CAOs remain undected near the ice edge if a high level is chosen.
- L111: What do you mean by the "climatological analysis"?
- L112: Please clarify which MCAO index you are refering to, i.e., indicate the level.

- L148: What do you mean by rasterize?
- L165ff: Please properly define the MCAO indices $m_i$
- Section 3.1: I really like the 3D figures! However, I believe the benefit of these does not yet come out clearly in the text. What specific features can only be made visible in 3D? I am sure there are, but the text remains vague about this. Also in principle, the "topography" of the isentropic surface used to delimit the MCAO air can easily be visualized in 2D maps. Please be specific about what can be gained from the 3D analysis that is not possible with more conventional charts?
- L303: Definition of new MCAO index: I don't understand the rationale behind choosing a fixed surface pressure $p_0$. You state that the new index "measures the vertical extent". In that case you should choose the actual surface pressure instead of a fixed value (see also Terpstra et al. 2021). This will make a big difference in regions where the MCAO air-mass is still shallow (i.e., near the ice edge).
- L330ff: Please clarify how you define the areas inside and outside the PLs. I understand from what has been said earlier that the area inside is the area within the PLs' radius. How then is the area outside of PLs delimted?
- L350: How are the pseudo-events defined spatially? Are you using the same areas as for the actual events?
- L366: 95[th] percentile with respect to space or time?
- L381: Are these the same pseudo-events as the ones before and why then do you explain their definition again? Or are they different? If so, I don't see how come. Please clarify.
- Figure 7: What is the step-size for the critical thresholds used?
- L457: Honestly, I am lost in this paragraph. I understand the authors want to find an alternative metric in order to avoid vertical interpolation to find the critical level where potential skin temperature equals potential temperature. The description of the methodology how to obtain this alternative index is more than confusing. I strongly suggest the authors rewrite this section entirely for better clarity. Furthermore, I believe that also some evidence needs to be shown that this approach is equivalent to the previous.

**Technical corrections:**

L9: remove comma after that

L22: … representing conditions conducive for …

L62: here and throughout the manuscript: please fix references by removing parentheses around year, should be (REF1 YYYY1; REF2 YYYY2…)

L84: remove full stop

L147: Barents and Nordic Seas

L157: than → as

L188: here and elsewhere: section should not be capitalized if it is not refering to a specific section

L202: here and elsewhere: add a comma before "e.g."

L263: remove sea

L286: Potential Vorticity **Unit**

L326: please remove parentheses around refs

L367: approximately

L384: Pseudo-events

L416: … has an area under the ROC curve (AUC score) …

L474: ROC analysis

L521: Terpstra et al. (2020) suggest

L551: front shear → forward shear

L560: (EDI; Wulff and Domeisen, 2019)

Caption Fig. 1: No need to repeat the detailed steps (1-3) here since you do that in the main text.

**References:**

Terpstra, A., Renfrew, I. A., & Sergeev, D. E. (2021). Characteristics of Cold-Air Outbreak Events and Associated Polar Mesoscale Cyclogenesis over the North Atlantic Region, *Journal of Climate*, *34*(11), 4567-4584

---

## Referee Comment (RC2)

**Review of "Interactive 3-D visual analysis of ERA 5 data: improving diagnostic indices for marine cold air outbreaks" by Meher et al.**

General comments:

The authors have represented marine cold air outbreaks (MCAO) in three dimensions using the software Met.3D and, using this viewpoint, have defined two new indices, one for MCAO and one for polar lows, which they claim give a more accurate association with polar lows than the previous versions of MCAO indices that depend on a subjectively chosen pressure level. Studying MCAOs in 3-D and their association with polar lows is interesting and new, to my knowledge, and worth being published in WCD. However, to my opinion, the manuscript needs major rewriting and rearranging. The authors should choose where to fully define the indices, either in the method or in the sections where they appear (if the authors want to follow the "workflow"). I found the manuscript hard to follow (especially Section 3.3.3). Some parts of the data and method section felt like a technical report or user guide and not like a scientific article. Moreover, I do not see the point in emphasizing the "workflow" in the first part of the manuscript. These issues are detailed in the specific comments below along with some others, such as some missing explanations/justifications that if added can make the manuscript clearer.

Specific comments:

Introduction:
Line 34: does it mean that machines that don't have GPUs cannot run 3-D software?
Lines 51-52: another index worth mentioning is the difference between the 500-hPa temperature and the sea surface temperature (used for example in Zappa et al. (2014) and the references therein).
Lines 76-77 and lines 77-80: are those two questions pertinent? The answers seem obvious to me: first use a software allowing 3-D visualization, which the authors do, and second yes it can.

Methodology:
The authors point out many times throughout the manuscript that they define and follow a workflow to create new indices for MCAO and polar lows. But isn't that the regular way to reach such a goal? First, study the particular cases and then find a generalization. I suggest to reduce the text about this "workflow" in the whole manuscript. Moreover, I don't see the purpose of Fig.1 and its caption because it is only repeating what is already written in the manuscript. Section 2.3, lines 189-190 can also be removed.
Lines 100-103, 108-116, 125-127, 133-137, 181-185: Why should the reader care about these aspects? I suggest to remove.
Lines 123-124: Why do the authors need to do this comparison of ERA5? Have they found a difference? If not, remove this sentence.
Line 124: What was the original grid of the downloaded data? Why not downloading ERA5 directly on the regular longitude/latitude grid?
Lines 148-151: Rewrite more simply. What does "rasterize" mean?

Line 152: Not clear. Where/Why is this comparison done? I suggest to remove this sentence.
Lines 165-166: Not everybody is expected to know what a "Receiver Operating Characteristic (ROC) curve and accuracy score" are. Can the authors explain what they are?
Lines 167-180: These lines refer to variables (m_theta, m_p, m_tr) that are defined very much further down the manuscript (in Section 3.2). Therefore, this part is confusing.
Line 171: ±12 hours seem long compared to a polar low lifetime. Can the authors justify this choice?
Line 173: Is it really useful to insert the Heaviside function that renders the equation more complicated? Clearly writing which timesteps are considered in this average should be enough.

Results:
Lines 194-195: "MCAOs are resolved in ERA5": why did the authors expect otherwise? MCAOs are a quite large-scale phenomenon.
Lines 197-198 and 272-273: Can the authors give the reasons for the increased MCAO vertical extent when moving south?
Lines 206-208: Remove sentence.
Line 212: Terpstra et al. (2016) used the STARS database and thus seems more appropriate here than Michel et al. (2018).
Line 238: The authors mention the calculation of "other variants of the conventional MCAO index". First, this is very vague, what are those indices? Second, I believe these indices are not shown in the paper, so I would remove this sentence or rephrase it.
Line 273: Please explain what the "conceptual descriptions" are.
Lines 307, 341: Shouldn't $p_0$ be equal to 1013.25 hPa (instead of 1013.15) that is the standard pressure?
Lines 345-347: Where can the reader see this result? Does this conclusion come from visual inspection? Please precise.
Section 3.3.3 is not very understandable to me and would benefit from a good rewriting. Here are some unclear aspects: What are the critical values of Mi used as thresholds, how are they spread over the "observed" values? Is the "observed binary score of PL occurrence" the same as "the binary empirical data about PL occurrence"? The "observed binary score of PL occurrence" is not a score but rather a mask. Why $M_i$ is "continuous"? It seems to be one map for each polar low track (number i). Remove sentence lines 403-404. On line 407, it is written $M_i>0$ but it should be $M_i=1$ because there are no other positive values and it would be consistent with equation 6. What are the "computing sensitivity and specificity for each threshold"? They seem to be associated with Youden's index but it is only mentioned later (and not even explained). What is the "accuracy score"? Is it the same as Youden's index? Line 434 mentions "a particular time" but $M_i$ is the average over several timesteps, isn't it? From the values given on line 449, the Youden's statistic should be equal to 0.78+0.58-1=0.36 which is pretty bad since the perfect value is 1 but the authors still seem to emphasize its good performance. A table recapping all values would be great (specificity, sensitivity, or just Youden's index and accuracy).
For clarity, I suggest to move Section 3.2.1 into Section 3.1 with Section 3.2 focusing only on the new indices (3.2.1 would be the new MCAO index, 3.2.2 about the new polar low index, 3.2.3 about the evaluation of the new indices and their association with polar lows, 3.2.4 about the region-specific index). To emphasize the usefulness of the new index, I think the authors should perform a composite of the new MCAO index for the polar low dates (as a

map) with the actual positions of the polar lows superimposed (such as the tracks shown in Terpstra et al. (2016) or only the genesis points with the 32 uncaught polar lows in another color).

Conclusions:
Lines 521-522: The sentence "Terpstra,,, initiation" is not clear, please rephrase. Does it mean that a MCAO that lasts for a long time does not promote more polar lows than a short-duration MCAO?
Lines 523-526: Can't polar low genesis happen both inside and at the outer edge of a MCAO? Moreover, the comparison between this study and previous studies might not be fair since the present study tends to average the index over the polar low duration when previous studies focused on the polar low genesis time.
Lines 531-539: The authors highlight the less computationally expensive 2-D new MCAO index but still advise to perform a first empirical study to determine the critical pressure, hence losing the advantage of the new index. Can't the authors give the best critical pressure to use in future studies?

Figure 3: What are the "typical 2-D wind-patterns during PLs" the authors refer to? Terpstra et al. (2016) seems a more appropriate reference here as it dealt with the STARS polar lows. On line 205, it is written that the polar low "formed east of the coast of Novaya Zemlya during the 19-20$^{th}$ December 2002". However, on the inlay of panel (a), the polar low is located west of Novaya Zemlya. Can the authors clarify? Could it be relevant to mention if the two polar lows selected formed in a forward or reverse shear environment?

Figure 5: The polar low location does not seem to move from one panel to the other whereas the time is not the same. The same comment applies to Movies 4 and 5. Is that normal?
Caption: the sentences "Movie 6 … in Fig. 2." "We count … (see Fig. 6 and 7)." seem both out of place here.

Figure 6: The authors wrote "the time of the PL on the 24$^{th}$ of March, 2011". What is the timestep?
For panels (b,e,f), there are no polar lows in the domain but is there any cold air outbreak?
Can the authors provide an error bar for the pseudo-events hit rate? For example, the authors could randomly draw 132 timesteps, count the number of matches and repeat those steps 1000 times to get a distribution of hit rates.
One may wonder what the MCAO looks like (in 3-D) when the MCAO cannot be associated with a polar low (e.g., one of the 32 polar low cases missed by the index in Fig. 6f).
How is the area of pseudo-events defined (used in panels c,f,i)?

The authors may consider changing the title of the manuscript to include the polar lows as estimating favorable environments for their formation is the end goal of the study if I am not mistaken.

Technical corrections (non-exhaustive list):

Line 22: infrastructure -> infrastructures
Line 65: differs -> differ
Line 147: Sea -> Seas
Line 197: Fig 2a-c -> Fig. 2c-e
Line 200: underline -> underlines
Lines 220-225: (3.2.1) -> (Section 3.2.1) and so on for 3.2.2, 3.2.3, and 3.2.4
Line 242: decreases -> decrease
Line 247: Fig. 4 -> Fig. 4a-c
Lines 247-248: "The magnitude of …pressure level" is a repetition of the sentence on line 241 so consider changing one of the two sentences.
Line 260: Fig.4 -> Fig. 4d-f
Line 287: removes the [] around the unit.
Line 316: at 2 PVU -> by the 2 PVU surface
Line 331: How are the within and outside areas defined?
Line 333: Fig. 6e -> Fig. 6b
Lines 334, 336: Fig. 6b,h -> Fig. 6e,h
Line 345: Fig. 6 -> Fig. 6h
Line 346: remove the word percent
Line 356: location -> locations
Line 385: remove "for details,"
Line 423: Youdens -> Youden's
Line 526: develop -> developing
Line 550: front -> forward
Figure 1: to be removed?
Figure 2: consider removing Fig, 2b. Is it correct that the 257-K isentrope is shown on the left panels (c-e) and the 267-K isentrope on the right panels (f-h)? It is not very clear. The contours on the cross-sections are not very visible. The caption should be simplified. Nordic Sea -> Nordic Seas. Remove [] around kg/kg. Remove "Fig. 3 illustrates … depicted here."
Figure 3: Maybe remove the dotted lines between the inlay and the 3-D picture for clarity.
Figure 4: (d-f) Vertical profile -> (d-f) Vertical profiles
Figure 6 panel (g): shouldn't it be 5th percentile instead of 95th for the polar low index?
Figures' caption: Does the description of the figures belongs to the caption or to the text? For example, in Fig. 2: "Before the start of the MCAO in case 1 (c), the cold air is located above sea-ice; it is then transported … over warmer oceans (d-e)." or in Fig. 3 "no symmetric slow wind eye is observed in the lower troposphere…further south."

References:
Terpstra, A., C. Michel, and T. Spengler, 2016: Forward and reverse shear environments during polar low genesis over the northeast Atlantic. Mon. Wea. Rev., 144, 1341–1354, https://doi.org/10.1175/MWR-D-15-0314.1.
Zappa, G., L. Shaffrey, and K. Hodges, 2014: Can polar lows be objectively identified and tracked in the ECMWF operational analysis and the ERA-Interim reanalysis? Mon. Wea. Rev., 142, 2596–2608, https://doi.org/10.1175/MWR-D-14-00064.1.

---

## Author Response (AR1)

July 2nd, 2021

**Response letter to the review of the manuscript**

*Title: Interactive 3-D visual analysis of ERA 5 data: improving diagnostic indices for marine cold air outbreaks and polar lows*

Authors: Meyer, M., Polkova, I., Modali, K. R., Schaffer, L., Baehr, J., Olbrich, S., Rautenhaus, M.

Dear Reviewers,

Dear Co-Editor,

we would like to thank both reviewers for their very detailed, constructive, and encouraging to positive reviews. Both reviewers have expressed concerns about the organization and writing of some parts of the manuscript, most notably the methods section and the last two subsections of the results section. We have conducted a major revision of the manuscript to address all concerns and recommendations. This includes moving the definition of indices into the methods section, reducing technical details in the methods section, improving the clarity of the last two subsections in the results section, an additional table to summarize performance metrics, additional discussions, and clarifications in different parts of the manuscript. Aside from the revision of the manuscript, we have conducted an additional composite analysis for comparing long-term average index values with all PL tracks, which is summarized in an additional figure in the revised manuscript. We have also added an appendix to cover additional details (examples of pseudo-events, examples of index maps during missed cases, tables to clarify the ROC analysis, numerical evidence for the comparability of the new MCAO index with the corresponding region-specific index).

Overall, we feel that the manuscript has greatly benefitted from the review. Our answers to each of the reviewer's comments (black font colour) are provided below in red font colour. All key comments have been addressed in the revised manuscript. The references to lines and paragraphs in our responses below refer to the revised manuscript without tracked changes.

Sincerely,

Marcel Meyer, Iuliia Polkova, Marc Rautenhaus, on behalf of all co-authors.

July 2$^{nd}$, 2021

Response to Reviewer #1 (RC1)

We gratefully acknowledge the constructive, encouraging, and very detailed feedback. The comments, critical remarks, and questions have been very helpful for improving the manuscript.

RC1 - Introductory Comment:

*This study investigates the usefulness of conventional marine cold air outbreak (MCAO) indices, which are typically based on the sea-air potential temperature difference, to identify regions where polar lows (PLs) develop and seeks to improve them in terms of their skill in predicting favourable regions for PL occurrence. For that purpose, the study advocates the use of 3D visualizations of MCAOs and PLs, informing the formulation of alternative indices. Based on case study analyses, two indices are then presented that perform particularly well in terms of predicting PL occurrence as demonstrated for a set of PLs from the STARS database and randomly chosen reference samples. The first index measures the depth of the unstable MCAO air mass by identifying the uppermost convectively unstable level, the MCAO top height, thereby following other studies (e.g., Terpstra et al. 2021). The second index also takes the role of upper-level forcing into account by considering the distance of the MCAO top height and the tropopause, which to my knowledge is novel. While the the presentation of figures is of high quality and, especially the 3D ones, are simply beautiful to look at, the writing of the manuscript and, in particular, also its organization need improvement. Also I am a bit unsure about novelty and scope of the study, as detailed below. Despite of that I think this study has the potential to become a valuable contribution to WCD if the manuscript is appropriately revised.*

Response:
- We appreciate the evaluation of the reviewer that the study can be a valuable contribution to WCD if the writing and organization of the manuscript is revised.
Action:
- We conducted a major revision of the manuscript. Details are given below.

RC 1 - General comments:

*RC1: One of the difficulties I have with this manuscript is whether it fits into the scope of WCD. On one hand, it introduces a workflow for 3D visualization of meteorological fields using the Met.3D software. While I have no doubt that 3D exploration of meteorological phenomena is highly useful and inspiring for formulating hypotheses, introducing such a workflow does not fall into the scope of WCD and other journals might be more appropriate (e.g., Geoscientific Model Development). On the other hand, the specific application, namely identifying better indices for capturing regions favourable for PL development, definitely falls into the scope of WCD. The way it is presented now, the paper tends towards both directions. I strongly suggest the authors focus the paper more towards the meteorological aspectes and less towards a 3D visualization workflow to make it more appropriate for WCD (of course keep the beatiful figures!).*

Response:
- We much appreciate that the use of 3-D exploration of meteorological phenomena for formulating/testing hypothesis was well received by the reviewer. Our author team is interdisciplinary with expertise in both atmospheric science and computer science, and besides the paper's major atmospheric focus on MCAOs and PLs, one of the objectives of our collaboration was to demonstrate how beneficial interactive 3-D visual analysis can be for the study of atmospheric phenomena. We hence intentionally decided to submit our manuscript to WCD to specifically reach the dynamic meteorology audience, for which we believe our approach will be interesting and beneficial. In our revision of the manuscript, we emphasize this point: the paper's focus is on the MCAO and PL indices, however, our approach to obtain the presented results included the 3-D visual analysis as a major part. For example, Met.3D enabled us to rapidly examine the issues with the conventional meteorological index for MCAOs and explore possible other ways of diagnosing MCAOs and PLs. By highlighting these

aspects in the paper, we provide WCD readers with a perspective on recent innovations in 3-D visualization research that demonstrates their use for exploration and hypothesis testing and how they can extend the toolset of conventional analysis methods. Since our methodology with Met.3D and ERA-5 data is entirely open-source and open-data, it will be straightforward to adapt to other kinds of atmospheric studies – this is where we believe we can create impact on the methods side. Given the recent availability of the full ERA-5 dataset, we also believe this is a timely aspect to be published.

Action:

- We have put additional weight on the meteorological aspects in the revised manuscript by:
    - improving the clarity of the aims, objectives, and motivation of our study (see, e.g., revised abstract, re-structured introduction and last paragraph in the introduction; revised manuscript),
    - conducting an additional composite analysis of index values and adding a corresponding new figure (Fig. 6, first paragraph, Sect. 3.3.3., revised manuscript),
    - re-structuring the methods (see Sect. 2, revised manuscript), including the removal of the more technical aspects of the methods (partly deleted, partly moved to Appendix A, revised manuscript),
    - adding additional details about meteorological aspects (see, e.g., definition of pseudo-events in Sect. 2.4. and Appendix B; discussion of index values during missed cases, Sect. 3.3.2; see also Appendix C and D with more detail about the classification schemes for evaluating indices; revised manuscript).

*RC1: I also have some reservations regarding the novelty of the presented work. Other studies (see Papritz and Spengler 2017 and Terpstra et al. 2021) have considered the depth of the unstable layer for identifying MCAOs before. In particular, Terpstra et al. (2021) also explore its usefulness for predicting PL occurrence. In addition, while the second index, which also takes the tropopause level into account, is – to my knowledge – novel, it bears some resemblance to the criterion developed by Kolstad (2011). What are the differences and advantages of your index? Why another index?*

Response:

- We were motivated in our study to address the subjective element in the choice of the pressure levels in various MCAO/PL indices and explore the vertical structure of MCAOs for a better association of MCAOs with PLs - something that to the best of our knowledge had not been discussed in the literature before. For this purpose, we conceptualize and evaluate new, alternative diagnostic indices.
- We present the application of novel methods for interactive 3-D visual data analysis to investigate MCAOs and PLs.
- With respect to the novelty of the new MCAO index, please note that our new MCAO index was developed independently of the work described by Terpstra et al. (2021). During the time of the study and at the time of submission of the initial manuscript we were not aware and could not have been aware of the work by Terpstra et al. (2021), because it was published after we had submitted the initial manuscript. While the metric introduced by Terpstra et al. (2021) is indeed very similar to the one we suggest here, the type of analysis is very different. In our opinion the two studies complement each other well.
- The other study that is mentioned in the comment of the Reviewer, the work of Papritz and Spengler (2017), does consider in some regards the depth of MCAOs, however focusses mainly on a Lagrangian trajectory analysis and does not suggest any metric that resembles the one we introduce here, and hence does not diminish the novelty of our diagnostic method.
- The new PL index is introduced here to specify and improve with more detail the index suggested by Kolstad (2011). In the initial and revised manuscript, we reference the work of Kolstad and state this clearly. The new PL index introduced here has the key advantage that it contains more information than the previous binary index and allows for a direct physical interpretation as the distance between the upper boundary of the MCAO and the dynamical tropopause. Further, we compute the index without using any empirical data, whereas in Kolstad (2011) the computation depends on the location of observed PLs.

Action:

- We improve clarity on the aims and objectives of our study (see re-phrasing of abstract and introduction, including a clear statement of objectives in the last paragraph of the introduction in the revised manuscript) and have included a reference and short discussion about the recent Terpstra et al. (2021) paper (see line 542ff, end of the 3ʳᵈ paragraph in the conclusion; revised manuscript).

*RC1: The manuscript is partly not very clearly organized, this applies especially to the methods section, which at great length explains the workflow including many technical and often irrelevant details (such as the supercomputer used) and unnecessary statements à la "We have implemented a functionality in Met.3D…" but omits important information about the methodology. Some unclear points are:*

- *Definition of the MCAO indices: these indices should be defined in the methodology section and not in the main part of the paper.*
- *Definition of "pseudo-events": How are the spatial regions chosen?*
- *For assessing the skill of the MCAO indices it is critical that in the reference sample no PLs occurred. How did you check that during the random time steps there are really no PLs? As I understand the STARS database contains only selected PL cases in the region but by far not all. For example, Michel et al. (2018) find about 243 PLs per winter season, compared to only about 13 PLs per winter in STARS.*
- *How are the 95th percentiles computed? Are they spatial or temporal? If they are spatial, what regions are considered?*

Response:
- Technical details: we feel that some of the perceived lack in clarity and inappropriateness in prioritization of contents (e.g. "irrelevant" details vs "important" information) may be attributed to different norms of writing in different communities, and the fact that this manuscript attempts to bridge the gap between meteorological (MCAOs, PLs) and methodological research (computer science methods for interactive 3-D visual data analysis). Some of the authors develop and maintain this software as their main line of research, and we think that including some of these details is relevant, because the "technical" and "scientific" spheres are intertwined and also interactive 3D data exploration is one of the scientific methods that we would like to promote in this study.
  - Action: we have shortened the technicalities in the revised manuscript (see revised Methods section). Some of the technical details have been put into the Appendix. We strongly reduce the use of the word "workflow" throughout the manuscript.

- Definition of MCAO indices: the structure of the initial manuscript was chosen to match the structure of the workflow that started with interactive visual data exploration. This was changed in the revised manuscript.
  - Action: We have re-arranged such that the definition of indices is covered in the methods section and have re-phrased this part to match the revised structure (see Methods section 2.3, revised manuscript).

- Definition of pseudo-events: these are defined to match the spatial scale (average track length and average radius) of actual PLs.
  - Action: Details of the definition of pseudo-events are given in the methods section of the revised manuscript (see Sect. 2.4, second paragraph; and Appendix B, revised manuscript).

- Assessing the skill of MCAO indices: we agree that we cannot be entirely certain that no other PL happened in addition to those covered by STARS during the time-interval covered by STARS. However, also the suggested paper by Michel et al. (2018) does not help in this respect as the authors in this paper report 243 "Polar Mesoscale Cyclones" (PMCs) and not 243 PLs. As Polar Lows are a sub-set of particularly intense PMCs, these numbers cannot be directly compared to those from STARS. Another observational dataset that focusses specifically on Polar Lows is described in Rojo (2019). They report 263 Polar Lows during a time-interval of 20 years. This corresponds to approximately 13 Polar Lows per year, which matches very well with the frequency of Polar Lows reported in STARS (13). We therefore assume that we cover key Polar Low events. However, of course, we cannot exclude the possibility that

we have missed single events. As we pick > 100 randomly chosen pseudo-events, an individual missed case would not have a notable effect on conclusions. A core focus of our study is comparing the existing indices and suggesting their improvement. As we use consistent datasets (ERA5 and STARS) for this comparison, we believe that our conclusions hold. Moreover, in the discussion of the manuscript we mention that a potential future line of work could include additional testing using more comprehensive PL data or other regions.

o 95[th] percentiles: the percentiles are spatial. We compute one time-averaged index value for each grid-cell during each event. Then the 95[th] percentile of all index values in the geographical domain of interest are considered.

o Action: explanation is given in the revised manuscript (see line 359ff, Sect. 3.3.2, 2[nd] paragraph, revised manuscript).

RC1: *The new MCAO index is evidently better suited for identifying regions favourable for PL development than conventional MCAO indices. However, it also has a clear disadvantage: it does no longer take the coldness of the air mass relative to the sea surface into account. Thus, it looses the useful property of conventional indices that they are directly proportional to the sea-air sensible heat flux (cf. Papritz et al. 2015). Hence, it is a matter of application which index is better suited. I would wish the authors include a discussion of this aspect.*

Response:
- We agree that the suitability of a particular index depends on the objective of the application and mention this explicitly in the discussion of the revised manuscript, including the disadvantage of not covering the surface heat fluxes.

Action:
- Added a short discussion about the suitability of different diagnostics for different objectives, including the reference mentioned by the reviewer (see line 564ff, 5[th] paragraph in the conclusions, revised manuscript).

**C1 - Specific comments:**

RC1 - L57: *please explain the term potential skin temperature, this may be unfamiliar to nonexperts*

Response:
- Yes, agreed.

Action:
- We have included a short description in the introduction of the revised manuscript (see introduction, 2[nd] paragraph, line 40, revised manuscript).

RC1 - L64ff, *discussion of different MCAO indices: I miss two important aspects here. First of all, CAOs are convectively unstable air masses subject to strong vertical mixing. This becomes also quite evident in the selected profiels that you show in Fig 4, where the variation of potential temperature with height is small within the CAO air compared to the tropospheric lapse-rate further aloft. Accordingly, the potential temperature is fairly uniform in the vertical, strongly reducing the sensitivity to the choice of characteristic pressure level.*

Response:
- The vertical level at which the mixing "ends" (i.e. the upper boundary of the MCAO) varies substantially between different areas of a MCAO and amongst different MCAOs. This means that the choice of the characteristic pressure level can make a substantial difference. For example, if choosing 700 hPa or 500 hPa one will miss entirely shallow MCAOs that reach up to around 850 hPa, obtaining a negative instead of a positive MCAO index. Further, while being fairly uniform in some of the well-mixed layers, the characteristic vertical profile of potential temperature within CAOs is not perfectly uniform, but rather characterized by an initial decrease directly above the surface, followed by a fairly uniform layer in which potentially temperature slowly increases, and then a strong increase at higher altitudes above

the MCAO (see Fig. 4 here or also in, e.g., Gryschka, 2018, Fig. 5-12). Therefore, the magnitude of MCAO index values clearly depends on the choice of the characteristic pressure level, which makes it difficult to compare different studies or apply generic classifications into weak or strong MCAOs. A reduced sensitivity to the choice of the characteristic pressure level, as outlined by the Reviewer, is only given for small variations of the characteristic pressure level (but not for the entire range of values used in previous studies - from 900 to 500 hPa) and conditional on the characteristic pressure level being located inside a very well-mixed vertical layer of the MCAO (which depends on the type of MCAO and the location within the MCAO and will often not be the case). Our analysis confirms that the choice of the specific pressure level can make a substantial difference (e.g., Fig. 4, as well as Fig. 8, the effect of different pressure thresholds on classification performance).

Action:
- We add a sentence on small variations inside a particular layer of MCAOs, and that there may be a reduced sensitivity to the choice of the characteristic pressure level (see line 51ff, introduction section, 3$^{rd}$ paragraph, revised manuscript).

*RC1 - Second, there is a clear preference for choosing a lower level because CAO air masses tend to be quite shallow initially and their depth only grows with fetch from the ice edge due to convective overturning and the associated entrainment of air from aloft. Hence, CAOs remain undected near the ice edge if a high level is chosen.*

Response:
- To us it seems that the preference depends on the objective of the study and might also depend on regional differences. For example, if the objective of the study is to detect cold air outbreaks at lower altitudes, then of course one should choose air temperature at a lower pressure level. However, one key disadvantage of choosing a lower level is that it tends to lead to the identification of very large areas with positive MCAO values very frequently, and it is thus difficult to relate the MCAO index to any useful information, e.g., regarding the risk for occurrence of Polar Lows. We show in our study that choosing some intermediate pressure level is best if one wants to maximize the link to observed PLs (see e.g., ROC analysis in Fig. 7 of the initial manuscript). Overall, we suggest that consideration of the vertical extent (as captured in the new MCAO index), as well as the vertical distance to the dynamical tropopause (as captured in the new PL index) can provide additional useful information. We do not mean to say that the new metrics are generically better for all kinds of studies. The choice of metric depends on the objective of the study.

Action:

- We discuss other metrics and state in the conclusion of the revised manuscript that we think the choice of the appropriate metric should depend on study-specific objectives (see conclusion section, 5$^{th}$ paragraph, revised manuscript).

*RC1 - L111: What do you mean by the "climatological analysis"?*

Response:
- We refer to the statistical analysis of all PLs and pseudo-events, which covers a time-interval of 10 years

Action:
- We rephrased this part in the revised version (see lines 122ff, Sect. 2.2, 2$^{nd}$ paragraph, revised manuscript).

*RC1 - L112: Please clarify which MCAO index you are refering to, i.e., indicate the level.*

Response:
- Yes, thanks.

Action:
- Has been clarified (see line 124, Sect. 2.2, 2$^{nd}$ paragraph, revised manuscript).

*RC - L148: What do you mean by rasterize?*

Response:
- We mean constructing a 2-D grid based on the point/track data in STARS ("rasterize" is often used in computer science).

Action:
- Re-phrased for clarification (see Sect. 2.4., 1st paragraph, revised manuscript).

*RC1 - L165ff: Please properly define the MCAO indices mi*

Response:
- In the initial manuscript, these were properly defined, but at a later point in the manuscript.

Action:
- We have re-arranged the methods section of the manuscript, such that all indices are defined in the methods section (see Sect. 2.3. for the definition of indices) and provide a reference to the Eqs. (see line 209, Sect. 2.4., 4th paragraph, revised manuscript).

*RC1 - Section 3.1: I really like the 3D figures! However, I believe the benefit of these does not yet come out clearly in the text. What specific features can only be made visible in 3D? I am sure there are, but the text remains vague about this. Also in principle, the "topography" of the isentropic surface used to delimit the MCAO air can easily be visualized in 2D maps. Please be specific about what can be gained from the 3D analysis that is not possible with more conventional charts?*

Response:
- Thanks, this is very encouraging. Some of the key advantages of 3-D visualizations are: (i) more comprehensive insights into the evolution of a meteorological phenomenon. This helps gaining an intuitive understanding of system dependencies and meteorological processes that are inherently 3D; (ii) ease of building mental/conceptual models when exploring comprehensive climate data-sets; (iii) avoid having to interpret numerous 2D images (e.g. at numerous vertical heights); and (iv) in particular, being able to summarize and communicate more information/data. We kept the figures in the manuscript as simple as possible to not overload the reader, but in principle, one could add various other elements to the 3D scenes (e.g., additional iso-surfaces, fronts, trajectories). A concrete example of the usefulness of the 3-D perspective is exploring the overlap of the dynamical tropopause and the upper boundary of MCAOs (Fig. 4, revised manuscript). The 3-D interactive analysis allows for simultaneous analyses of various iso-surfaces, which helps for exploring the relation between MCAOs and dynamical tropopause further, instead of having to perform additional computations and plotting various 2D charts. One of the core ideas behind providing additional 3-D figures and Movies is to provide an example case for the application of novel methods which can be applied generically to all sorts of meteorological phenomena (rather than focussing too much on the specific details and features). Aside from the 3-D perspective, the essential advantage of the method is the capacity for interactive data exploration, i.e. being able to use a graphical user-interface to slide cross-sections through data-volumes or varying iso-values on-the-fly (see movies 1,2). We think that the combination of these two aspects, 3-D perspectives and interactively moving through data scenes, can provide very valuable complements to more conventional 2D charts. We see particular advantage for 3-D interactive visualization during the initial explorative phase of scientific workflows, for detailed case-studies, and for conceptualizing and testing hypotheses about 3-D dynamics of meteorological phenomena.

Action:
- We extend the introduction by more detail about potential benefits of 3D IVA, including some additional references (see 4th paragraph, introduction, revised manuscript). Also, we summarize aims of the 3D analysis in the results section of the revised manuscript (see first paragraph, Sect. 3.1., revised manuscript) and re-phrase the corresponding paragraph in the conclusion (2nd paragraph, conclusions, revised manuscript).

July 2nd, 2021

*RC1 - L303: Definition of new MCAO index: I don't understand the rationale behind choosing a fixed surface pressure p0. You state that the new index "measures the vertical extent". In that case you should choose the actual surface pressure instead of a fixed value (see also Terpstra et al. 2021). This will make a big difference in regions where the MCAO air-mass is still shallow (i.e., near the ice edge).*

Response:
- During our analyses we have tried both variants, using a fixed constant or the time-dependent surface pressure. The main rationale for choosing a fixed constant is simplicity, as well as relevance for the key questions addressed in this study. As stated by the reviewer, using the actual surface pressure may make a difference in those areas where the MCAO air-mass is still shallow – if meteorological conditions are characterized by substantial deviations from the standard surface pressure. However, the focus in this study is on areas with high MCAO depth, and in these areas the choice between a fixed constant and the actual surface pressure does not make a substantial difference and may therefore be neglected. For example, if we assume that surface pressure varies on average by around 10-20 hPa and consider that the average MCAO height inside of the area of PLs is around 300 hPa (see text), then this corresponds to approximately 5%, which can be neglected for the sake of simplicity and reducing the amount of required input-data to the necessary minimum. Of course, if one were to measure the vertical extent of MCAOs near the sea-ice as the main research objective, then the actual surface pressure should be used instead.

Action:
- We mention in the revised manuscript that this is an approximation (see lines 152 and 158, Sect. 2.3, 2nd and 3rd paragraph, revised manuscript).

*RC1 - L330ff: Please clarify how you define the areas inside and outside the PLs. I understand from what has been said earlier that the area inside is the area within the PLs' radius. How then is the area outside of PLs delimted?*

Response:
- The area inside and outside of PLs is defined based on the observational data in STARS. All grid-cells within the PLs radius around the PLs track are defined as "inside". All grid-cells that are not within the PLs radius around the PLs track, but over open waters, are defined as outside of PLs. Note also that we only consider areas with lower-level instability when calculating the new index variants, hence the cut-off at zero in the distribution of index values in Fig. 5, panels (b,e,h).

Action:
- Added a sentence about the definition of inside/outside areas to the main text (see line 325, 1st paragraph, Sect 3.3.1., revised manuscript).

*RC1 - L350: How are the pseudo-events defined spatially? Are you using the same areas as for the actual events?*

Response:
- The pseudo-events are defined to match the average spatial scale, w.r.t. to track length and radius, of the actual events reported in STARS. In the revised manuscript we provide more details on the definition.

Action:
- Added details of the definition to the main text (see Sect. 2.4., revised manuscript).

*RC1 - L366: 95th percentile with respect to space or time?*

Response:

- With respect to space. The index values are time-averaged for each event and we then compute the 95$^{th}$ percentile with respect to space

Action:
- Clarified in the revised manuscript (see Sect. 3.3.2., 2$^{nd}$ paragraph, revised manuscript).

*RC1 - L381: Are these the same pseudo-events as the ones before and why then do you explain their definition again? Or are they different? If so, I don't see how come. Please clarify.*

Response:
- Yes, these are the same ones as defined before.

Action:
- More details of the definition are given in the Methods section (see Sect. 2.4, 2$^{nd}$ paragraph, revised manuscript).

*RC1 - Figure 7: What is the step-size for the critical thresholds used?*

Response:
- Critical thresholds are chosen to cover the range of all index values with a step-size depending on the index variant. We defined 20+1 thresholds to cover the range of index values with a constant step-size. For example, values for the new MCAO index are chosen from the interval [0,600] with a fixed step-size of 30, i.e. we test the thresholds 0, 30, 60, 90, …, 600.

Action:
- We clarify this in the revised manuscript (see Sect. 3.3.3., second paragraph, revised manuscript).

*RC1 - L457: Honestly, I am lost in this paragraph. I understand the authors want to find an alternative metric in order to avoid vertical interpolation to find the critical level where potential skin temperature equals potential temperature. The description of the methodology how to obtain this alternative index is more than confusing. I strongly suggest the authors rewrite this section entirely for better clarity. Furthermore, I believe that also some evidence needs to be shown that this approach is equivalent to the previous.*

Response:

- The method described in this section provides a way for determining the characteristic pressure level in the conventional MCAO index from observational data. The resulting metric, termed here the regional MCAO index, has the advantage that it is computationally cheaper while maintaining the same relation to observed PLs as the new MCAO index with a given critical threshold for classification.
- We have substantially re-phrased and re-written the Section with an emphasis on the opening paragraph. We have added a numerical example to the Appendix for illustrating that both index variants with their corresponding thresholds lead to the same dichotomized index values and hence the same skill for their association with PLs. In ongoing work (not shown), we have additionally tested and confirmed that both index variants (with their corresponding thresholds) yield the same prediction skill in a subseasonal prediction system.

Action:

- We conducted a re-write of parts of this Section, and provide additional numerical evidence in the Appendix (see Sect. 3.3.4 and Appendix D, revised manuscript).

*RC1: Technical corrections:*

- *L9: remove comma after that*
  - Response: Thanks, corrected.
- *L22: … representing conditions conducive for …*
  - Response: Thanks, corrected.
- *L62: here and throughout the manuscript: please fix references by removing parentheses around year, should be (REF1 YYYY1; REF2 YYYY2…)*
  - Response: Thanks, corrected. We went through the manuscript to ensure consistency.
- *L84: remove full stop*
  - Response: Thanks, corrected.
- *L147: Barents and Nordic Seas*
  - Response: Thanks, corrected.
- *L157: than → as*
  - Response: Thanks, rephrased the entire paragraph.
- *L188: here and elsewhere: section should not be capitalized if it is not referring to a specific section*
  - Response: Adapted.
- *L202: here and elsewhere: add a comma before "e.g."*
  - Response: Thanks, corrected.
- *L263: remove sea*
  - Response: Thanks, corrected.
- *L286: Potential Vorticity Unit*
  - Response: Adapted.
- *L326: please remove parentheses around refs*
  - Response: Thanks, corrected.
- *L367: approximately*
  - Response: Thanks, corrected.
- *L384: Pseudo-events*
  - Response: Thanks, corrected.
- *L416: … has an area under the ROC curve (AUC score) …*
  - Response: Thanks, corrected.
- *L474: ROC analysis*
  - Response: Thanks, corrected.
- *L521: Terpstra et al. (2020) suggest*
  - Response: Thanks, corrected.
- *L551: front shear → forward shear*
  - Response: Thanks, corrected.
- *L560: (EDI; Wulff and Domeisen, 2019)*
  - Response: Thanks, corrected.
- *Caption Fig. 1: No need to repeat the detailed steps (1-3) here since you do that in the main text.*
  - Response: adapted by removing Fig. 1 entirely.

*RC1 - References:*

- *Terpstra, A., Renfrew, I. A., & Sergeev, D. E. (2021). Characteristics of Cold-Air Outbreak Events and Associated Polar Mesoscale Cyclogenesis over the North Atlantic Region, Journal of Climate, 34(11), 4567-4584*
- Response: references included in the revised manuscript.

July 2nd, 2021

**Response to Reviewer #2 (RC2)**

We thank the reviewer for the positive, constructive, and very detailed feedback. The comments, questions and recommendations have been very helpful in improving the manuscript.

*RC2 - General comments:*
*The authors have represented marine cold air outbreaks (MCAO) in three dimensions using the software Met.3D and, using this viewpoint, have defined two new indices, one for MCAO and one for polar lows, which they claim give a more accurate association with polar lows than the previous versions of MCAO indices that depend on a subjectively chosen pressure level. Studying MCAOs in 3-D and their association with polar lows is interesting and new, to my knowledge, and worth being published in WCD.*
Response:

- We greatly appreciate the evaluation of our work as interesting, new, and worthy of publishing in WCD.

*RC2 - General comments:*
*However, to my opinion, the manuscript needs major rewriting and rearranging. The authors should choose where to fully define the indices, either in the method or in the sections where they appear (if the authors want to follow the "workflow"). I found the manuscript hard to follow (especially Section 3.3.3). Some parts of the data and method section felt like a technical report or user guide and not like a scientific article. Moreover, I do not see the point in emphasizing the "workflow" in the first part of the manuscript. These issues are detailed in the specific comments below along with some others, such as some missing explanations/justifications that if added can make the manuscript clearer.*
Response:

- We are grateful for the detailed comments on improving clarity and structure of the manuscript. We have carried out a major rewrite to address these comments in the revised manuscript. On reflection, we feel that some of the perceived lack in clarity and inappropriate style of writing (e.g., "technical report" vs. "scientific article") of the initial manuscript can be attributed to the fact that we attempted to bridge the gap between meteorological (MCAOs, PLs) and methodological research (computer science methods for interactive 3-D visual data analysis). In parts, different norms of writing (e.g., w.r.t. prioritization of technical details) exist in these different communities. One of the aims of the paper is demonstration of the benefit of interactive 3-D visual analysis, as available in Met.3D, for interactive visual analysis of MCAOs and PLs. Due to this somewhat interdisciplinary approach, we felt that it is important to also include some basic technical details. However, considering the comments of the reviewer and the core scope of WCD, we have re-structured and re-phrased to improve clarity and adapted the writing style.

Action:

- Re-writing and re-arranging of the manuscript (details below).

RC2 - Specific comments:

*RC2: Introduction:*
*RC2 - Line 34: does it mean that machines that don't have GPUs cannot run 3-D software?*
Response:

- Not 3-D software in general, but Met.3D as a specific framework requires GPUs to achieve its interactive performance. In general, computation of 3-D features, such as a 3-D isosurface, does not require a GPU. However, the methods for interactive 3-D visual data analysis that we demonstrate in this paper are computationally demanding. To achieve interactive performance, a single image needs to be generated within a few milliseconds in order to achieve interactive frame rates (typically > 25 images per second) that facilitate smooth interaction with the visualization. Met.3D in particular makes use of features only available in GPUs that support the OpenGL application programming interface in version 4.3 or higher,

so a workstation requires such graphics hardware to run the software. More information about the computational expense of visualization algorithms such as 3-D isosurfaces raycasting used to visualize the potential temperature isosurfaces is available in the Met.3D publication (Rautenhaus et al., 2015).

Action:
- Re-write of this part of the introduction (see Introduction, 4th paragraph, revised manuscript).

*RC2 - Lines 51-52: another index worth mentioning is the difference between the 500-hPa temperature and the sea surface temperature (used for example in Zappa et al. (2014) and the references therein).*

Response:
- Yes, thanks.

Action:
- We included the additional reference (see line 42, 2nd paragraph in the introduction, revised manuscript).

*RC2 - Lines 76-77 and lines 77-80: are those two questions pertinent? The answers seem obvious to me: first use a software allowing 3-D visualization, which the authors do, and second yes it can.*

Response:
- We agree that the questions were phrased a bit too generically. The first question was meant to ask if we can use Met.3D for interactive visual analysis of ERA5 and if so, which steps / methods are required to do so. The second question aimed at exploring the level of detail to which 3D features are resolved.

Action:
- We re-phrased the paragraph, including adaptation of one question as a statement and specification of the other question (see last paragraph of the Introduction, revised manuscript).

*RC2: Methodology:*

*RC2 - The authors point out many times throughout the manuscript that they define and follow a workflow to create new indices for MCAO and polar lows. But isn't that the regular way to reach such a goal? First, study the particular cases and then find a generalization. I suggest to reduce the text about this "workflow" in the whole manuscript. Moreover, I don't see the purpose of Fig.1 and its caption because it is only repeating what is already written in the manuscript. Section 2.3, lines 189-190 can also be removed.*

Response:
- We mean to advocate a workflow that includes interactive 3-D visual analysis. The emphasis here is on the interactive 3-D visual data analysis, rather than the logical approach for reasoning (from a particular case to generalization). One of the aims of the manuscript is showcasing, by example, that interactive visual data analysis can be a useful complement to standard workflows that do not include interactive 3-D visual analysis. We have revised the manuscript to point this out more clearly and we agree that the word "workflow" has been used too often.
- The purpose of Fig. 1 was providing a graphical overview of the core steps of the analysis. However, we agree that it is not an essential figure and have removed it from the manuscript.

Action:
- Revisions to emphasize interactive visual data analysis instead of a rather generic focus on workflows (see, e.g., abstract, 4th paragraph introduction, first paragraph of the Methods Section; the use of the word "workflow" has been substantially reduced and Fig. 1 has been removed).

*RC2 - Lines 100-103, 108-116, 125-127, 133-137, 181-185: Why should the reader care about these aspects? I suggest to remove.*

Response:
- Lines 100-103: the main line of research of some of the authors evolves around developing and applying these methods (Met.3D has been developed in and is maintained by our research group). In the manuscript, we want to briefly summarize these novel methods that we make available to the meteorological community. We hope that it can be an advantage for some readers to get an

idea of the various kinds of methods that are available in the visualization/computer graphics community, so that s/he can think about what may be useful for her/his own research. The development of new methods and their application forms one central part of the study presented here, hence we think some details about these methods should be covered in the methods section.

- Lines 108-116: this is a brief description of the selection of MCAO/PL cases and the methods for interactive 3-D visual data analysis that we have used. To the best of our understanding a description of the methods (for interactive visual data analysis) we use for parts of the study, along with a description of the selection of cases, is a standard and relevant detail for the methods section of a manuscript describing the interactive visual analysis of selected cases, among others.
- Lines 125-127 and 133-137: one important ongoing research question at the interface of high-performance computing and meteorology is how to bring together technical aspects of scientific work (e.g., computing hardware) with domain-science. These two areas are intertwined and often cannot be separated clearly. We therefore felt that it may be relevant to include a short summary of the technical setup that forms the basis for the research described here. This is particularly so, because the manuscript is interdisciplinary at the interface of computer science and meteorology. With the details provided, the reader has the necessary information for evaluating if the novel method could be useful for his/her work. Without these details, this is not possible. However, some of this information one can obtain from the previous papers on Met.3D, we therefore have shortened this paragraph to address the reviewer's suggestion.
- Line 181-185: the information was provided to give an idea of the computational requirements for conducting the type of analysis described in this paper.

Action:
- We have addressed the comments by re-arranging, and re-phrasing of the methods Section, also including deletion and moving of more technical aspects to the Appendix (see Methods Section and Appendix A, revised manuscript).

*RC2 - Lines 123-124: Why do the authors need to do this comparison of ERA5? Have they found a difference? If not, remove this sentence.*
Response:
- This was just to check and confirm consistency.

Action:
- The sentence was removed.

*RC2 - Line 124: What was the original grid of the downloaded data? Why not downloading ERA5 directly on the regular longitude/latitude grid?*
Response:
- The ERA5 data are natively stored on different grids (spectral and Gaussian) at the ECMWF - depending on the variable. When downloading on regular lat/lon grid, the conversion is done automatically behind the scenes. We use the data (which is a copy of the archive at ECMWF) hosted on DKRZ's supercomputer "Mistral" and perform the computationally intensive parts of the analysis on Mistral, because it is much quicker to perform the analysis on the computer, where the data is stored instead of downloading it all.

*RC2 - Lines 148-151: Rewrite more simply. What does "rasterize" mean?*
Response:
- Agreed that simplifying would be better. By rasterize we mean constructing a 2-D grid based on the point/track data in STARS.

Action:
- Re-phrased to clarify (see lines 179ff, Sect. 2.4., 1$^{st}$ paragraph, revised manuscript).

*RC2 - Line 152: Not clear. Where/Why is this comparison done? I suggest to remove this sentence.*
Response:

- We assess the ability of the indices to serve as proxies for PLs by comparing them with empirical data on PLs. This sentence was meant to describe the format of the empirical data (a 2-D matrix) used for this comparison.

Action:

- Re-phrased and re-arranged the methods section so that this is clarified in the revised manuscript (see Sect. 2.4., revised manuscript).

*RC2 - Lines 165-166: Not everybody is expected to know what a "Receiver Operating Characteristic (ROC) curve and accuracy score" are. Can the authors explain what they are?*

Response:

- Agreed, we were not sure what should be assumed to be known and what should not.

Action:

- Added a short summary and additional references for details (see Methods Section 2.4., 3$^{rd}$ paragraph, lines 203ff, and Results Section 3.3.3, 3$^{rd}$ paragraph, lines 420ff revised manuscript).

*RC2 - Lines 167-180: These lines refer to variables (m_theta, m_p, m_tr) that are defined very much further down the manuscript (in Section 3.2). Therefore, this part is confusing.*

Response:

- In the initial manuscript, the definition of the indices followed the IVA of individual cases, as we wanted to emphasize that IVA can be helpful for generating new perspectives (metrics).

Action:

- We have re-structured the manuscript to include the definition of indices in the methods part (see Sect. 2.3., revised manuscript).

*RC2 - Line 171: ±12 hours seem long compared to a polar low lifetime. Can the authors justify this choice?*

Response:

- The PL tracks in STARS are derived from satellite images and do not cover the full development/genesis time. By starting 12 hours before the detected onset, we allow for typical spin-up times (comparable approach also used in Terpstra, 2016). The idea is to capture atmospheric conditions around the times of observed PLs with an additional time-buffer to account for genesis, decay and measurement uncertainties in STARS when determining the exact timing.

Action:

- We add a short explanation and reference to the revised manuscript (see lines 213f, Sec. 2.4, revised manuscript).

*RC2 - Line 173: Is it really useful to insert the Heaviside function that renders the equation more complicated? Clearly writing which timesteps are considered in this average should be enough.*

Response:

- Agreed.

Action:

- Adapted notation / phrasing (see line 215ff, Sect. 2.4., 4$^{th}$ paragraph, revised manuscript).

*RC2 - Results:*
*RC2 - Lines 194-195: "MCAOs are resolved in ERA5": why did the authors expect otherwise? MCAOs are a quite large-scale phenomenon.*

Response:

- We were curious to see how well 3-D features are resolved, considering especially the high vertical resolution of ERA5 data on model levels. To our knowledge, nobody has looked at MCAOs and PLs

in 3-D in ERA5 before and so we simply wanted to briefly confirm the expectation that these phenomena are resolved, followed by examples and details given in that section.

Action:

- We deleted that part (new beginning of the second paragraph in Sect. 3.1., revised manuscript).

RC2 - Lines 197-198 and 272-273: Can the authors give the reasons for the increased MCAO vertical extent when moving south?

Response:

- Agreed, that in 272-273, initial manuscript, we could have provided reasons in addition to the reference containing some of the typical reasons.

Action:

- We added a short description of dominant processes in addition to the reference (see lines 296ff, Sect. 3.2.2, last paragraph, revised manuscript).

RC2 - Lines 206-208: Remove sentence.

Response:

- We describe in this sentence one of the specific advantages of interactive 3D data analysis. The other Reviewer has suggested to outline in more detail the advantages of 3-D interactive analysis. Therefore, we think it is more appropriate to keep this sentence.

RC2 - Line 212: Terpstra et al. (2016) used the STARS database and thus seems more appropriate here than Michel et al. (2018).

Response:

- We refer to a specific figure in the Michel et al. paper (Fig. 8), which shows a directly related 2-D perspective. In this respect we think the reference that we provided is more appropriate. However, we add the additional references as well

Action:

- We added a reference to Figure 8 in Michel et al, 2018, and we add the additional Terpstra et al. (2016) reference suggested by the reviewer (see Sect. 3.1., last paragraph, revised manuscript).

RC2 - Line 238: The authors mention the calculation of "other variants of the conventional MCAO index". First, this is very vague, what are those indices? Second, I believe these indices are not shown in the paper, so I would remove this sentence or rephrase it.

Response:

- We have provided the specific references to the concrete other indices that we have implemented (see references at the end of the sentence in the initial and revised manuscript), hence we are not sure why this is described as "very vague" by the reviewer. We want to avoid explicitly writing down the exact definition of these additional indices, as it would fill up too much space.

Action:

- We clarified in the text that the provided references refer to the specific indices (see lines 146f, Section 2.3., 1$^{st}$ paragraph, revised manuscript).

RC2 - Line 273: Please explain what the "conceptual descriptions" are.

Response:

- Agreed.

Action:

- We added a short explanation (see Sect. 3.2.2., last paragraph, revised manuscript).

RC2 - Lines 307, 341: Shouldn't p_0 be equal to 1013.25 hPa (instead of 1013.15) that is the standard pressure?

Response:

- Yes, thank you, these were typos in the manuscript. We used 1013.25 for the calculations.

Action:
- Corrected (see lines 162 and 334, revised manuscript).

*RC2 - Lines 345-347: Where can the reader see this result? Does this conclusion come from visual inspection? Please precise.*

Response:
- As indicated by the reference to Figure 5 (revised manuscript), the reader can see parts of these results in the Figure. Panels b, e, h, show the distribution of index values during all PL events in different spatial domains for the different index variants. One could use visual inspection to see that the mean of the blue curve in Panel h is approx. 345 hPa, as described in the sentence referenced by the reviewer. However, in our analysis, the mean value and the frequency of intrusion of the upper-level anomaly into the lower-level instability is calculated in a separate analysis.

Action:
- Clarified phrasing (see line 340, 3[rd] paragraph in Sect. 3.3.1, revised manuscript).

*RC2 - Section 3.3.3 is not very understandable to me and would benefit from a good rewriting.*
*RC2 - Here are some unclear aspects:*
*RC2 - What are the critical values of Mi used as thresholds, how are they spread over the "observed" values?*

Response:
- The critical values are chosen such that they cover the entire range of index values. For example, the new MCAO index, $m_p$, takes on values in [0,600] and we define 21 threshold values, evenly distributed in that interval. This yields a binary index that is compared with the observed PL tracks. On re-reading, we realize that we should not have chosen the phrase "interval of all observed values" in line 412 of the initial manuscript, as it can be confused with the empirically observed PL values.

Action:
- We have re-phrased and added an example for clarifying this in the revised manuscript (see Sect. 3.3.3., 2[nd] paragraph, revised manuscript).

*RC2 - Is the "observed binary score of PL occurrence" the same as "the binary empirical data about PL occurrence"? The "observed binary score of PL occurrence" is not a score but rather a mask.*

Response:
- Yes, it is the same. Sorry about the confusion.

Action:
- Re-phrasing (see second paragraph, Sect. 3.3.3., revised manuscript).

*RC2 - Why $M_i$ is "continuous"? It seems to be one map for each polar low track (number i).*

Response:
- We meant that the index values are continuous variables (i.e., real number). There is one map for each index (denoted by i) during all events (i.e., the 132 polar lows and the 132 pseudo-events).

Action:
- Re-phrasing (Sect. 3.3.3., 2[nd] paragraph, revised manuscript).

*RC2 - Remove sentence lines 403-404.*

Response & Action:
- We re-phrased the entire paragraph, incl. this sentence.

*RC2 - On line 407, it is written $M_i>0$ but it should be $M_i=1$ because there are no other positive values and it would be consistent with equation 6.*

Response & Action:
- Yes, thanks, corrected (see line 417, Sect. 3.3.3., revised manuscript).

*RC2 - What are the "computing sensitivity and specificity for each threshold"? They seem to be associated with Youden's index but it is only mentioned later (and not even explained). What is the "accuracy score"? Is it the same as Youden's index?*

Response:

- The ROC curves are obtained by computing sensitivity and specificity for each threshold and plotting sensitivity (true positive) against 1 – specificity (false positive) (see Fig. 7). Youden's index is defined as sensitivity + specificity -1. The accuracy is defined as (True positives + True negatives) / N.

Action:

- We added the definition of the metrics and references for details to the text (see Methods Section 2.4, 3$^{rd}$ paragraph, revised manuscript).

*RC2 - Line 434 mentions "a particular time" but M_i is the average over several timesteps, isn't it?*

Response:

- Yes, here we refer to the time of the event (and not to the hourly time-steps in ERA5). Events are defined via the time and/or the location of PLs, as described in the text. The indices M_i are averaged over several time-steps of the ERA5 data to obtain one index map per event (for all PL and all pseudo-events).

Action:

- Re-phrased and added two tables for summarizing the classification scheme (see Sect. 3.3.3. and Appendix E, revised manuscript).

*RC2 - From the values given on line 449, the Youden's statistic should be equal to 0.78+0.58-1=0.36 which is pretty bad since the perfect value is 1 but the authors still seem to emphasize its good performance.*

Response:

- In the discussion of the initial (and revised) manuscript we explicitly state that the complex genesis of PLs cannot be fully captured with simple indices such as the conventional or the new indices introduced here (line 545 of the initial manuscript). We want to underline that the performance of the new index is better than the old index in our experimental setting.

*RC2 - A table recapping all values would be great (specificity, sensitivity, or just Youden's index and accuracy).*

Response:

- Yes, thanks, we agree that a summary of key metrics could be helpful. However, the specificity and sensitivity values, as well as Youden's index, can be obtained directly from Figure 7 for each index and all thresholds (sensitivity=true positive rate; specificity = 1- false positive rate; Youden's index=sensitivity + specificity – 1). To avoid redundancy, and, because Youden's index is just used to identify the best critical threshold, which is then used to calculate the accuracy, we do not summarize these metrics in the table, but focus on the key metrics described in the text: AUC values (area under the curve) and accuracy scores for the best classifier are summarized in a separate table.

Action:

- Added an additional table to summarize the performance metrics (see Table 2, revised manuscript).

*RC2 - For clarity, I suggest to move Section 3.2.1 into Section 3.1 with Section 3.2 focusing only on the new indices (3.2.1 would be the new MCAO index, 3.2.2 about the new polar low index, 3.2.3 about the evaluation of the new indices and their association with polar lows, 3.2.4 about the region-specific index).*

Response:

- Section 3.2.1 in the initial manuscript is entitled "sensitivity of conventional MCAO indices to the choice of the characteristic pressure level" and Section 3.1 is entitled "interactive 3-dimensional visual analysis of MCAOs and PLs in ERA5". To us it makes little sense to merge these Sections as they cover different lines of analysis.

Action:

- We moved the indicator definitions into the methods section. The results Section in the revised manuscript is also partially re-arranged. We order the results Section according to the type of

analysis (IVA, statistical analysis), allowing for a comparative description of different indicators, and re-name some of the headings for clarity.

*RC2 - To emphasize the usefulness of the new index, I think the authors should perform a composite of the new MCAO index for the polar low dates (as a map) with the actual positions of the polar lows superimposed (such as the tracks shown in Terpstra et al. (2016) or only the genesis points with the 32 uncaught polar lows in another color).*

Response:
- We conducted the suggested composite analysis and added the corresponding maps to the revised manuscript. The composites for the old and the new MCAO index provide additional support for the usefulness of the new index. However, by calculating the long-term average, one loses a lot of information compared with the far more detailed statistical analysis based on the ROC curves and accuracy values, because the latter are analysed on a per-event basis for all PL events and pseudo-events. To further emphasize the usefulness of the new index for classifying/distinguishing PL events from no-events, we have computed the difference of the long-term averaged index values between PL and non-PL events (pseudo-events) (see Fig. 6 and first paragraph of Sect. 3.3.3., revised manuscript). This is more informative than the mean during all PL events as it shows that, on average, there are larger differences in index values for the new MCAO index (compared with the conventional one) between times of observed PLs and times with no PLs (pseudo-events) in those areas where most PLs have been reported. These larger differences indicate that a classification into PL and non-PL events based on index values will be more successful for the new index, compared with the conventional MCAO index. The composite analysis further supports the systematic performance assessment based on the metrics (ROC curves, AUC, accuracy) described in the manuscript. The new index is suited better than the conventional one for indicating where and when PLs occur. As stated above, it is important to note that we focus on comparing the different indices, showing that the new index is better relative to the conventional index for being a PL proxy. Clearly, also the new metrics analysed here cannot fully capture the complex genesis of PLs, but only provide an indication (this is explicitly noted in the discussion of the initial and revised manuscript).

Action:
- Conducted additional composite analysis (see Fig. 6 and first paragraph of Sect. 3.3.3., revised manuscript).

*RC2 - Conclusions:*
*RC2 - Lines 521-522: The sentence "Terpstra,,, initiation" is not clear, please rephrase. Does it mean that a MCAO that lasts for a long time does not promote more polar lows than a short-duration MCAO?*
Response:
- The previous study of Terpstra appears to indicate that the duration of MCAOs is not relevant for polar mesoscale cyclogenesis, of which PLs are a sub-class.

Action:
- Re-phrased (see Conclusion, 3[rd] paragraph, revised manuscript).

*RC2 - Lines 523-526: Can't polar low genesis happen both inside and at the outer edge of a MCAO? Moreover, the comparison between this study and previous studies might not be fair since the present study tends to average the index over the polar low duration when previous studies focused on the polar low genesis time.*
Response:
- Yes, polar low genesis can happen both inside and at the outer edge of MCAOs. However, again, one should keep in mind how the MCAO is defined. Please note that this was described in the initial manuscript (see the sentence after the one referenced above, i.e., starting in line 524 of the initial manuscript). We point out that there are remaining open questions and differences in previous work. The aim of this paragraph is a general discussion of results in the context of previous work, rather than a specific quantitative comparison. Hence, we think it is appropriate and interesting to mention some of the previous work even if the experimental setup is not identical. Also, it is not

the case that all of the previous studies mentioned in that discussion focus on genesis only (see e.g., Kolstad, 2011).

*RC2 - Lines 531-539: The authors highlight the less computationally expensive 2-D new MCAO index but still advise to perform a first empirical study to determine the critical pressure, hence losing the advantage of the new index. Can't the authors give the best critical pressure to use in future studies?*
Response:
- The choice of the appropriate indicator depends on the specific study purpose. In some situations, the conventional index might be sufficient (e.g. interest is on surface heat fluxes), in other situations the new MCAO index might be more suitable (e.g. when looking for a simple metric to indicate when and where PLs might occur), and in other situations the region-specific index might be better (e.g. when focusing in detail on a specific region). One of the disadvantages of the conventional MCAO index is the necessity for subjectively choosing a characteristic pressure level. We suggest here a method on how this pressure level can be determined from empirical data of PLs in a specific region. Whilst this method does rely on performing one computationally expensive analysis (though feasible on most hardware), this does not mean that the advantage of the region-specific index is lost, because once the critical pressure level is determined from observational data it can be used in various subsequent analyses/computations. For example, one could imagine the following use-case: conduct one first study to determine the critical pressure level in the MCAO index for a particular region (e.g. the Barents sea) and then use the region-specific MCAO index with the parameter "critical pressure level" obtained from available empirical data for computing risks in operational risk forecast settings (which rely on computationally cheap metrics for quick computations). The critical pressure value depends on region-specific details, as, e.g., the orography, meteorological processes specific for the region and not least the availability of observational data about PLs. We suggest assessing the best critical pressure level on a regional basis, instead of extrapolating the one determined here for the Barents and the Nordic Seas.

*Comments to figures*

*RC2 - Figure 3: What are the "typical 2-D wind-patterns during PLs" the authors refer to? Terpstra et al. (2016) seems a more appropriate reference here as it dealt with the STARS polar lows.*
Response:
- We refer to a specific figure (Fig. 8) in the (Michel et al., 2018) paper, which indicates, amongst others, that wind-speeds are on average low in the centre / eye of PLs. The long-term average in Fig. 8 of Michel et al. (2018) shows similarities (visual inspection) to the single PL event shown here. Hence we chose the Michel et al. (2018) reference.
Action:
- Added the specific reference to Fig. 8 to emphasize what we refer to. Also, we added the Terpstra et al. reference suggested by the reviewer as an additional reference (see caption to Fig. 2, revised manuscript).

*RC2 - On line 205, it is written that the polar low "formed east of the coast of Novaya Zemlya during the 19-20th December 2002". However, on the inlay of panel (a), the polar low is located west of Novaya Zemlya. Can the authors clarify?*
Response:
- Thanks for pointing out this typo/wrong description. The PL formed west of Novaya Zemlya.
Action:
- Corrected.

*RC2 - Could it be relevant to mention if the two polar lows selected formed in a forward or reverse shear environment?*

Response:

- During our analysis, we have discussed whether to include additional analysis about forward and reverse shear environments. We have decided against it, mostly because the manuscript is already long enough, and our core focus is on the interface between interactive 3-D visual analysis of single cases and statistical analysis of simple indicators derived from the IVA, which do not distinguish between reverse and forward shear conditions. Hence, the distinction between forward and reverse-shear conditions for the two specific PLs illustrated in the Fig. is not directly related to our core arguments and scope of the manuscript, which is why do not include an analysis of this. By means of visual comparison of the 3D wind patterns in our fig with the averaged wind patterns on the 2D cross-section in Fig. 8 of the Michel et at (2018) paper, it appears that the first PL (panel a) with the symmetric low wind eye may have formed during reverse shear conditions, while the second PL (panel b) formed during forward shear conditions. For future work, it could be very interesting to investigate any potential differences in 3D patterns during forward and reverse shear conditions.

Action:

- Add note state similarities to reverse shear conditions for PL 1 (see caption Fig. 2, and line 250, revised manuscript)

*RC2 - Figure 5: The polar low location does not seem to move from one panel to the other whereas the time is not the same. The same comment applies to Movies 4 and 5. Is that normal?*
Response:

- The vertical pole is purposefully plotted at a fixed location to indicate the mean position of the observed PL. The key focus of the figure and movies is on the dynamics of 3-D iso-surfaces and index values. We keep the position of the PL fixed, to not overload the visualization with too many moving objects.

Action:

- Added a short note to the caption to clarify this (see caption to Fig. 4, revised manuscript).

*RC2 - Caption: the sentences "Movie 6 … in Fig. 2." "We count … (see Fig. 6 and 7)." seem both out of place here.*
Response:

- We had put these descriptions there to guide the reader from figure to figure. In our experience this can be helpful, as many readers do not look into all details of the text.

Action:

- We rephrase the reference to Movie 6 and keep this in the caption as it is directly related to the Figure. We deleted the cross-references to other Figures and Movies not directly related (see adapted caption to Fig. 4, revised manuscript).

*RC2 - Figure 6: The authors wrote "the time of the PL on the 24th of March, 2011". What is the timestep?*
Response:

- We show the time-averaged index values here. These are averaged over all time-steps around the observed PL event, as detailed in the methods Section. On re-reading, we realize that the word "time" was a bit confusing as we are showing the time-average.

Action:

- Re-phrased to avoid confusion about the time-step and add note to clarify these are time-averaged index values (see caption to Fig. 5, revised manuscript).

*RC2 - For panels (b,e,f), there are no polar lows in the domain but is there any cold air outbreak?*
Response:

- The panels (b,e,f) show the distribution of index values in different geographical domains, as given in the observational STARS data. For example, the distribution of index values in all grid-cells during all PL events inside the area of PLs reported in STARS. It was not analysed if there are any observational/empirical data about MCAOs in these areas (to our knowledge these kinds of observational data do not exist). However, panel b shows the distribution of the conventional

MCAO index in these areas. As this index is often used to define the existence of MCAOs, its distribution serves as an indication of cold air outbreaks in these areas.

*RC2 - Can the authors provide an error bar for the pseudo-events hit rate? For example, the authors could randomly draw 132 timesteps, count the number of matches and repeat those steps 1000 times to get a distribution of hit rates.*

Response:

- The times of the pseudo-events are randomly chosen (random number generator). For each pseudo-event, the required ERA5 data (including the 3D temperature and 3D potential vorticity fields) is obtained, and all index variants are computed for the duration of the pseudo-event (defined as the average duration of observed PL events in STARS). Repeating this analysis for 1000 different combinations of pseudo-events is beyond the scope of this study, as it would require substantial additional computational effort. Also, our experimental setup in other parts is based on considering the same number of pseudo-events as there are observed PL events in STARS. As mentioned in the discussion of the initial (and revised) manuscript, it could be an interesting line of future work to consider a more comprehensive dataset of observed PLs (e.g. Rojo, 2019), and then computing a full climatology, i.e. considering the index values for all days of all winters. The focus of our manuscript is on a workflow that connects the application of new methods for interactive 3-D visual data analysis to the conceptualization of new index values and a first statistical test of the performance of newly derived indices.

*RC2 - One may wonder what the MCAO looks like (in 3-D) when the MCAO cannot be associated with a polar low (e.g., one of the 32 polar low cases missed by the index in Fig. 6f).*

Response:

- There is a great variety of 3-D shapes of MCAOs both in cases with and without PLs. We focus here on two simple aspects of the 3-D structure, namely the height of MCAOs and the distance to the dynamical tropopause. These aspects are analysed in detail for cases with and without PLs. As shown, there is a tendency for MCAOs to be higher at times and in areas where PLs occur. Hence, MCAOs are, on average, a bit shallower when the MCAO cannot be associated with a polar low. However, picking a single example that has some form of representative character for times when PLs are missed is very difficult to impossible, due to the immense diversity of shapes. We do not see the advantage of showing an arbitrary example in 3-D. Instead, to address the comment and provide more insights into the cases when the indices do not capture the PLs, we add additional figures to the Appendix, which show index values overlayed with the position of observed PLs for two examples of events when PLs are missed.

Action:

- We added a note about index maps during missed cases (see lines 353f, Sect. 3.3.2, revised manuscript) and an additional figure to the Appendix for showing index values and observed PLs for one of the cases when the PL was not captured by diagnostics (see Appendix C, revised manuscript).

*RC2 - How is the area of pseudo-events defined (used in panels c,f,i)?*

Response:

- The area of pseudo-events is defined to match the spatial scales of observed PLs in STARS as follows: (i) we compute the mean track length and the mean radius of all PLs in STARS; (ii) we randomly select (random number generator) a starting location for the pseudo-event somewhere in the geographical domain of interest (over open water); (iii) we define a random track of the pseudo-PL event with the mean track length from observational data but with randomly chosen directions; (iv) we use the random track of the pseudo-event (with length corresponding to average length of all PLs in STARS) together with the average radius of all observed PLs to define the area of each PL event. We agree that in the initial manuscript we had provided too little detail about the method used to define pseudo-events (Sec. 2.4. of the initial manuscript).

Action:

- Added details of the definition of random pseudo-events to the methods Section (see second paragraph of Sec. 2.4., revised manuscript). Also, we added a figure to the Appendix for illustrating two examples of random pseudo-events (see Appendix B, revised manuscript).

*RC2 - The authors may consider changing the title of the manuscript to include the polar lows as estimating favorable environments for their formation is the end goal of the study if I am not mistaken.*
Response:
- We agree and have appended the title.

Action:
- Change title to "Interactive 3-D visual analysis of ERA 5 data: improving diagnostic indices for marine cold air outbreaks and polar lows".

*RC2 - Technical corrections (non-exhaustive list):*
- *RC2 - Line 22: infrastructure -> infrastructures*
  - Response: Adapted.
- *RC2 - Line 65: differs -> differ*
  - Response: Thanks, corrected.
- *RC2 - Line 147: Sea -> Seas*
  - Response: Thanks, corrected.
- *RC2 - Line 197: Fig 2a-c -> Fig. 2c-e*
  - Response: Thanks, corrected.
- *RC2 - Line 200: underline -> underlines*
  - Response: Thanks, corrected.
- *RC2 - Lines 220-225: (3.2.1) -> (Section 3.2.1) and so on for 3.2.2, 3.2.3, and 3.2.4*
  - Response: Thanks, corrected according to WCD guidelines, i.e. use of "Sect."
- *RC2 - Line 242: decreases -> decrease*
  - Response: Thanks, corrected.
- *RC2 - Line 247: Fig. 4 -> Fig. 4a-c*
  - Response: Adapted.
- *RC2 - Lines 247-248: "The magnitude of …pressure level" is a repetition of the sentence on line 241 so consider changing one of the two sentences.*
  - Response: Adapted.
- *RC2 - Line 260: Fig.4 -> Fig. 4d-f*
  - Response: Adapted.
- *RC2 - Line 287: removes the [] around the unit.*
  - Response: Thanks, corrected. We replaced [] with \unit{} to match WCD submission guidelines.
- *RC2 - Line 316: at 2 PVU -> by the 2 PVU surface*
  - Response: Thanks, corrected.
- *RC2 - Line 331: How are the within and outside areas defined?*
  - Response: According to track and radius given in STARS. Added a note in the text to specify this.
- *RC2 - Line 333: Fig. 6e -> Fig. 6b*
  - Response: Thanks, corrected.
- *RC2 - Lines 334, 336: Fig. 6b,h -> Fig. 6e,h*
  - Response: Thanks, corrected.
- *RC2 - Line 345: Fig. 6 -> Fig. 6h*
  - Response: Adapted.
- *RC2 - Line 346: remove the word percent*
  - Response: Thanks, corrected.
- *RC2 - Line 356: location -> locations*

- o   Response: Instead, adapted both to plural for consistency with "cases".
- RC2 - Line 385: remove "for details,"
  - o   *Response: Thanks, corrected.*
- RC2 - Line 423: Youdens -> Youden's
  - o   Response: Thanks, corrected.
- RC2 - Line 526: develop -> developing
  - o   Response: Develop seems a more appropriate use.
- RC2 - Line 550: front -> forward
  - o   Response: Thanks, corrected.
- RC2 - Figure 1: to be removed?
  - o   Response: Removed.
- RC2 - Figure 2: consider removing Fig, 2b. Is it correct that the 257-K isentrope is shown on the left panels (c-e) and the 267-K isentrope on the right panels (f-h)? It is not very clear. The contours on the cross-sections are not very visible. The caption should be simplified.
  - o   Response:
    - ▪ Yes, the isentropes are correct, and we added a note to the caption to clarify which isentrope is shown in which panel.
    - ▪ The line contours are plotted to showcase this feature in Met.3D.
    - ▪ The caption was shortened.
    - ▪ We prefer to keep some details in the caption, e.g. references to text, or other figures and movies, to guide the reader through the manuscript.
- RC2 - Nordic-> Nordic Seas.
  - o   Response: Thanks, corrected.
- RC2 - Remove [] around kg/kg.
  - o   Response: Thanks, corrected. We now use \unit{}.
- RC2 - Remove "Fig. 3 illustrates … depicted here."
  - o   Response: Thanks, corrected.
- RC2 - Figure 3: Maybe remove the dotted lines between the inlay and the 3-D picture for clarity.
  - o   Response: We prefer to keep the dotted line for ease of reference between 3-D picture and inlay.
- RC2 - Figure 4: (d-f) Vertical profile -> (d-f) Vertical profiles
  - o   Response: Thanks, corrected.
- RC2 - Figure 6 panel (g): shouldn't it be 5th percentile instead of 95th for the polar low index?
  - o   Response: Thanks, corrected.
- RC2 - Figures' caption: Does the description of the figures belongs to the caption or to the text? For example, in Fig. 2: "Before the start of the MCAO in case 1 (c), the cold air is located above sea-ice; it is then transported … over warmer oceans (d-e)." or in Fig. 3 "no symmetric slow wind eye is observed in the lower troposphere…further south."
  - o   Response:  The caption was shortened. However, we prefer to keep some detail so that the Figure can be understood easily, and the reader can follow key aspects and navigate between Figures/Movies by just looking at the figures and captions.

*RC2 - References:*
- o   *Terpstra, A., C. Michel, and T. Spengler, 2016: Forward and reverse shear environments during polar low genesis over the northeast Atlantic. Mon. Wea. Rev., 144, 1341–1354,*
  - ▪ *https://doi.org/10.1175/MWR-D-15-0314.1.*
- o   *Zappa, G., L. Shaffrey, and K. Hodges, 2014: Can polar lows be objectively identified and tracked in the ECMWF operational analysis and the ERA-Interim reanalysis? Mon. Wea. Rev., 142, 2596–2608,*
  - *https://doi.org/10.1175/MWR-D-14-00064.1.*

- Response: References included in the revised manuscript.

July 2nd, 2021

---

## Editor Decision (ED1)

wcd-2021-20

Editor decision – comments to the authors

**Interactive 3-D visual analysis of ERA5 data: improving diagnostic indices for marine cold air outbreaks and polar lows**

by M. Meyer et al.

Dear Dr. Meyer

Many thanks for your revisions and for addressing the points raised by the reviewers in great detail. I am happy to accept your paper for publication in WCD subject to technical corrections, as suggested below. Congratulations and thank you for submitting your paper to WCD!

L75: provide → provided

L83: discuss → discussed

L125/127 and throughout the paper (e.g., also in caption of Figs. 1, 2, 3, etc.): the format of dates differs from case to case and is not according to journal standards. The format should be "on 18 December 2002", or "at XX:00 UTC on 18 December 2002".

L135: symbols t, pv, … are not defined. I don't think that they are required here, you can omit them. In case you decide to keep them, then please introduce the symbols and then t should be T, z should be Z, and pv should be PV.

L140: I think you have all these references already in the introduction (where they are important). Here they can be omitted.

Eq. 2: I am not sure that this notation is elegant and fully appropriate. In my understanding, $m^p\_\theta$ is a function of $p$. Therefore $m_\theta(p)$ would be, in my view, the more appropriate notation. The full equation should then read: $m\_\theta(p) = \theta_{skin} - \theta(p)$

L168: for the vertical levels → for all vertical levels

L183 and in many other places: during years → during the years

L209: should read "performance of the diagnostic …"

L214: should read "(in this case a diagnostic index)"

L219: confusing … what is $T$? Is $T = t_{end} - t_0$ ?

L222: should read "(a comparable approach … was used in Terpstra …"

Caption of Fig. 1: kg/kg → kg kg$^{-1}$

L257: m/s → m s$^{-1}$

L258: references should be in chronological order

L262: I don't understand "Resolved dynamics of cloud cover". But maybe it is anyway best to delete this sentence?

Caption of Fig. 2: please shorten this caption. The caption should only explain what the lines, colors, … in the plot mean, but the caption should not interpret the figures. The sentences "As expected …" and "The illustrated aspects …" can be deleted and the rest can be shortened.

L290-300: here I had the feeling that this is mainly repetition from Sect. 2.3. Please consider to shorten this part.

Eq. 5: again the notation, I suggest to write $\theta(p*)$ instead of the subscript.

L317: delete "defined here …", this was mentioned before.

L353: low or absent forcing → weak or no forcing

L361: delete "or any fitting procedure" (or explain what it means)

L391: "as compared" → "compared"

Eq. 7: and another notation: in the same spirit as above I suggest $m\_\theta(p_{crit}) = \theta_{skin} - \theta(p_{crit})$

L534: do you really want to emphasize the "slow-wind eye" here? In every circular symmetric vortex the wind speed must be zero at the center, so the "slow-wind eye" is not so much a surprise(?).

L546: report → reported

List of references: unlike the main paper, this part is not yet in very good shape. Often page numbers, volume numbers or DOIs are missing. Sometimes journal names are abbreviated,

sometimes not (please check in other WCD publications for the standard abbreviations to be used). Also please update the reference of the Afargan-Gerstman et al. paper.

I am looking forward to receiving the final version of your manuscript.
With best regards,
Heini Wernli

---

## Author Response (AR2)

02/08/2021

**Response letter to the review of the revised manuscript**

*Title: Interactive 3-D visual analysis of ERA 5 data: improving diagnostic indices for marine cold air outbreaks and polar lows*

Authors:

Meyer, M., Polkova, I., Modali, K. R., Schaffer, L., Baehr, J., Olbrich, S., Rautenhaus, M.

Dear Reviewers,

Dear Co-Editor,

we greatly appreciate the positive review of the revised manuscript. The suggestions of reviewers have helped further improving the manuscript. All comments have been addressed in this minor revision.

Our answers to each of the reviewer's comments (black font colour) are provided below in red font colour. The references to lines and paragraphs in our responses below refer to the revised manuscript without tracked changes. The revised manuscript with tracked changes shows all changes conducted as part of this minor revision (i.e., compared with the last submitted version of the manuscript).

Yours sincerely,

Marcel Meyer and Marc Rautenhaus, on behalf of all co-authors.

Response to Reviewer #1 (RC1)

Thank you very much for reviewing the revised manuscript. We gratefully acknowledge the constructive and positive review, which has helped further improving the manuscript.

RC1 - Introductory Comment:

*The authors put a lot of effort into the revision of the paper, whose presentation is now much improved, and they responded well to my earlier comments. The intention of the paper now comes out clearly and the methodology is very well explained. The paper provides an important step forward in terms of diagnostic indices for MCAOs and identifying regions favourable for PL development. And it is an excellent showcase for the potential of 3D interactive analysis in meteorology. Hence, I am convinced it provides a valuable contribution to WCD.*
*I have one main comment and a few minor comments, as detailed in the following [...].*

Response:
- We appreciate the positive evaluation and the additional suggestions for further improvements.
Action:
- We conducted a minor revision of the manuscript to address all comments. Details are given below.

RC1: *The collocation of the maximum MCAO depth and the occurrence of a PL is perhaps not too surprising given that PLs often require for their genesis some upper-level forcing in the form of a positive potential vorticity anomaly, as you outline in 3.2.3. Such upper-level positive potential vorticity anomalies are generally associated with an upward doming of the underlying isentropic surfaces (e.g., Hoskins et al. 1985) and, hence, also of the zero-isosurface m_theta_p, leading to reduced stability (and lifting). This relationship is the likely reason why the new MCAO index performs well in predicting PL occurrence; via the upward doming of the upper boundary of the MCAO, it contains implicit information on upper-level forcing. The conventional MCAO index does not contain such information and instead it is purely a measure of the coldness of the CAO air relative to the sea surface. The coldness of the air is likely not the most critical parameter for PL development as long as upper-level forcing is absent. I think the authors should discuss this aspect.*

Response:

- Yes, we agree, this is a good complement for the discussion. One of our initial thoughts behind the new Polar Low Index was that it could potentially capture this interplay of lower-level and upper-level anomalies even better than the new MCAO index, but as results indicate, it appears the simpler new MCAO index does already contain the implicit information about the upper-level forcing, as it performs just as well and even a bit better than the new Polar Low index.
Action:
- We add a discussion about the implicit information about upper-level forcing contained in the new MCAO index to the revised manuscript (see line 542f, third paragraph in the Conclusion, revised manuscript).

RC1 - L50: *MCOA -> MCAO*

Response & Action:

- Thanks, corrected.

*RC1 - L80: Even though 3D depiction of atmospheric fields is not the standard, meteorologist are well aware of the potential of 3D visualization since a long time, see for example Figs. 6.12 and 6.14 in Uccellini (1990).*

Response & Action:

- Yes, agreed and added to the introduction (see line 78-79, Introduction, revised manuscript).

RC1 - L102: *No need to mention the 37 standard pressure levels on which ERA5 output is available since you are anyway using data on model levels.*

Response & Action:

- Agreed and deleted.

RC1 - Caption Table 1*: Briefly mention in the caption the meaning of the different symbols (e.g., p_tr, p0, p\* etc...)*

Response & Action:

- Yes, included.

RC1 - L173: *Maybe rephrase as "... if the lower-level instability extends all the way to the tropopause." or similar. I don't think an MCAO induces an instability, but the instability is rather a defining characteristic of an MCAO.*

Response & Action:

- Re-phrased, thanks.

RC1 - L195: *Fig. B1 suggests that the lysis point is selected randomly within a circle with a radius corresponding to the mean track distance around the genesis point. Is this correct? The way it is written now, one may think that several such increments are computed*

Response:

- No, that's not correct. The tracks are generated incrementally. In the first step, we generate a random genesis location. In the second step, we randomly generate the direction for the PL. Subsequently, we generate a set of randomized track-increments. The path-direction chosen randomly at the beginning dominates the direction of all subsequent track-increments, but we allow for very small random variations around the initially chosen direction at each increment. This leads to almost straight track paths but does allow for some small variations. We also define a small random component on the length of each track-increment, such that the overall track length of all random control events corresponds to the observed mean track length of all PL events in STARS, but there are some small variations amongst individual pseudo-events. We chose this setup to include some variability in the randomly defined control-events, as a simple way for approximating some of the variability in the observed tracks.

Action:

- We adapted the phrasing (see line 194ff, revised manuscript) and replaced one of the examples in Fig. B1 with another example from our analysis for clarification.

RC1: *Suggest to remove "more realistic insights". I don't think one way or the other of depicting meteorological data is more realistic, but the 3D approach is certainly useful for an interactive exploration.*

Response:
- We agree, it is difficult to say what is more realistic in general and have removed this part of the sentence.

RC1 - L289: *delete "in"*

Response:
- Thanks, corrected.

References:

Hoskins, B. J., M. E. McIntyre, and A. W. Robertson, 1985: On the use and significance of isentropic potential vorticity maps.
Quart. J. Roy. Meteor. Soc., 111, 877–946, doi:10.1002/qj.49711147002.

Uccellini, L. W., 1990: Processes contributing to the rapid development of extratropical cyclones.
Extratropical Cyclones. The Erik Palmen Memorial Volume, C. W. Newton, and E. O. Holopainen, Eds., American Meteorological Society, Boston, USA, 81–105.

Response:
- Both references are included in the revised manuscript.

Response to Reviewer #2 (RC2)

Thank you very much for the positive and thorough review. We have addressed all comments in the revised manuscript.

RC2 - Line 40*: "sea surface potential temperature over the ocean": it seems redundant to have sea surface and ocean here. Suggestion: potential temperature calculated using the sea surface temperature.*

Response & Action:
- We adapted the phrasing to avoid appearing redundant (see line 40, revised manuscript).

**RC2 - Line 102 (and Appendix A):** *"137 vertical model levels, 37 pressure levels". I am confused here: what are you actually using in Met.3D, model levels or pressure levels? Has the vertical interpolation onto pressure level to be performed before using Met.3D or does Met.3D interactively perform the vertical interpolation? Moreover, in Appendix A, only horizontal remapping is mentioned. Therefore, I suggest to explicitly write about the vertical interpolation somewhere, either in Appendix A if the vertical interpolation is performed before using Met.3D or in Sect. 2.1 if the vertical interpolation is performed in Met.3D. The authors can also maybe explicitly write the 37 pressure levels available as well.*

Response:
- For all our key analyses we are using data on model levels. During initial testing and for the example in Fig. D1-b in the Appendix, we additionally used data on pressure levels. Met.3D can process both, data on pressure and on model levels, and the vertical interpolation is done on-the-fly.

Action:
- We removed the pressure levels from the text (see line 102f, revised manuscript) to avoid misunderstandings and added to Sec. 2.1 (line 107, revised manuscript) a note about internal on-the-fly vertical interpolation as part of Met.3D.

**RC2 - Line 105:** *I do not understand why a polar stereographic projection is required before using Met.3D (what does it do to the grid of the data?). Isn't it Met.3D itself that performs the projection at the plotting stage? If yes, I suggest a sentence like "Met.3D displays fields using a polar stereographic projection." or similar. Moreover, can the grid step in degrees be written here (for example 0.5ºx0.5º)?*

Response:
- Met.3D performs the projection of simple 2D line-geometries (such as country borders) at the plotting stage, but it does not perform the projection of the 3-D data variables at the plotting stage. That's because this would slow down the 3D visualization algorithms (e.g., for computing 3D iso-surfaces) too much for interactive visual analysis. Therefore, the projection is done in a pre-processing step using the climate data operators.

Action:
- We added the grid resolution to the text (see lines 105f, revised manuscript).

**RC2 - Line 120:** *3-D features -> the 3-D structure*

Response & Action:

- Yes, adapted phrasing (see line 121, revised manuscript).

**RC2 - Line 129:** *"on a large geographical domain (all longitudes; northern latitudes in the interval 25-90º" -> over the Northern Hemisphere (north of 25ºN)*

Response & Action:

- Yes, thanks, good point; we re-phrased (see line 130f, revised manuscript).

RC2- Lines 129-130: *I suggest to remove "grid-dimension in lat-lon height: 261x1441x137".*

Response & Action:
- Yes, deleted.

RC2 - Line 132: *I suggest to also remove "on a smaller grid (440x440x137)". The first "440" is not smaller than the "261" of line 130 so that is confusing. I suggest to replace "on a smaller grid….Nordic Seas" with "on a smaller domain covering the Barents and Nordic Seas". The polar stereographic aspect is already mentioned in Sect. 2.1.*

Response & Action:
- Yes, thanks, adapted to simplify the sentence (see line 131ff revised manuscript).

RC2 - Line 136: *I do not understand "at which air aloft is considered for calculation of the MCAO index". I suggest to remove as the rest of the sentence is clear without this part.*

Response:
- This was a reminder/repetition of the meaning of the characteristic pressure level, as the level at which potential temperature is taken for the calculation of conventional indices.

Action:
- We deleted this part of the sentence to avoid repetition and address the comment (see line 136-137, revised manuscript).

RC2 - Line 164: *"without lower-level instability": does it mean that the lowest-level potential temperature is larger than the potential skin temperature? If yes, maybe write it (with a formula).*

Response:
- We compare potential air temperature with potential skin temperature for a set of vertical levels, and if potential skin temperature is smaller than potential air-temperature for all vertical levels, including the bottom ones, than we define this as "no lower-level instability".

Action:
- We clarified the phrasing in the revised manuscript (see line 164, revised manuscript).

RC2 - Line 171: *remove "(in hPa)" as no units are written except in Table 1.*

Response & Action:
- Yes, adapted phrasing.

RC2 - Lines 214-216: *if the hours with m_p=0 are not considered in the temporal average, then I suggest to remove "and setting the index to zero for all other hours" and instead write that the index is equal to 0 if all time steps have m_p=0. Or is there something I have misunderstood?*

Response:
- You are correct, thanks, the phrasing contained a kind of duplication.

Action:
- We have adapted the phrasing (see line 216ff, revised manuscript).

RC2 - Lines 319, 394, and caption of Fig. 6: *I like the additional composites performed by the authors that I think really show the differences between the conventional and new MCAO indices. However, I do not think the term "long-term average" describes what is calculated. From my understanding, what is done here is a composite of the maps of the indices averaged over the polar lows duration.*

Response:
- For each single event (PL and pseudo-event), we average the index values, as described in the text. For the composite analysis we compute the average of index values during all events (PL and pseudo-events, respectively). We agree that the term "long-term average" could be interpreted as considering index values for all days of the time-period, which would be misleading.

Action:
- We rephrased the caption to Fig. 6 and the text around line 394.

RC2 - Section 3.3.2: *would it make sense to switch the 3rd ("The number…") and 4th ("To assess…") paragraphs and maybe merge the new 4th paragraph with the 5th ("Results…")? I think it would read better.*

Response:
- Yes, agreed.

Action:
- Re-structuring of parts of Section 3.3.2.

RC2 - Line 398: *to help the reader, maybe point at the area between Iceland and Norway where there are more yellow colours in Fig. 6c than in Fig. 6d -> (e.g., compare the values in the area between Iceland and Norway in Fig. 6c,d).*

Response:
- Yes, good point, thanks.

Action:
- We adapted the text (see line 398, revised manuscript).

RC2 - Line 421: *does a true positive rate of 1 imply that the index always correctly detect a PL (100% match)? If yes, could it be mentioned somewhere (main body or caption of Fig. 7) to more easily understand the ROC curves?*

Response:
- Yes, a true positive rate of 1 means that all PLs would be detected.

Action:
- We mention perfect true positive rate in the text (see line 424, revised manuscript).

Technical comments:

RC2 - Line 50: *MCOA -> MCAO*
Response: Thanks, corrected.

RC2 - Line 65: *"that maximizes the link to observed PLs" -> that maximizes this link.*
Response: Yes, thanks, adapted.

RC2 - 212: *shouldn't it be t\*start and t\*stop in the parenthesis (with the stars)?*
Response: Yes, thanks, corrected.

RC2 - Line 213: *end -> lysis*
Response: Ok, adapted.

RC2 - Lines 219, 302: *upper level -> upper-level*
Response: Yes, adapted.

RC2 - Line 239: *Movies 1-2 -> Movies 1, 2*
Response: Ok, adapted.

RC2 - Line 289: *remove one "in"*
Response: Thanks, corrected.

RC2 - Line 321: *remove comma after "areas"*
Response: Thanks, corrected.

RC2 - Line 338: *Fig. 5-h -> Fig. 5h*
Response: Thanks, corrected.

RC2 - Caption of Fig. 5: *"(>=95-th percentile)" is correct for the MCAO indices (panels c and f) but shouldn't it be the 5th percentile for the polar low index (panel i)?*
Response: Yes, it is, thanks. We have corrected the text in the caption accordingly.

RC2 - Line 404: *remove comma after "Sect. 2.4)"*
Response: Ok, adapted.

RC2 - Line 406: *Fig. 6-e -> Fig. 6e*
Response: Yes, adapted.

RC2 - Eq. 6: *columns (:) after ifs are not needed*
Response: Ok, simplified.

RC2 - Line 433: *Fig. 7 -> Fig. 7a*
Response: Yes, specified here and for 7b as well.

RC2 - Line 463: *add a full stop (.) at the end of the sentence after "Table 2)"*
Response: Thanks, corrected

RC2 - Line 576: *proof -> prove*
Response: Thanks, corrected.

---

## Author Response (AR3)

10/08/2021

**Response to the editor**

*Title: Interactive 3-D visual analysis of ERA 5 data: improving diagnostic indices for marine cold air outbreaks and polar lows*

Authors:

Meyer, M., Polkova, I., Modali, K. R., Schaffer, L., Baehr, J., Olbrich, S., Rautenhaus, M.

Dear Prof. Wernli,

thank you very much for handling and the manuscript and helping to improve it further by means of these final technical corrections. We greatly appreciate the positive decision and the opportunity to publish in WCD.

You find below a brief response to each of your comments. In summary: we have addressed all technical corrections as suggested.

Yours sincerely,

Marcel Meyer, on behalf of all co-authors.

10/08/2021

L75: provide → provided
Response: adapted.

L83: discuss → discussed
Response: adapted.

L125/127 and throughout the paper (e.g., also in caption of Figs. 1, 2, 3, etc.): the format of dates differs from case to case and is not according to journal standards. The format should be "on 18 December 2002", or "at XX:00 UTC on 18 December 2002".
Response: adapted throughout.

L135: symbols t, pv, … are not defined. I don't think that they are required here, you can omit them. In case you decide to keep them, then please introduce the symbols and then t should be T, z should be Z, and pv should be PV.
Response: deleted, thanks.

L140: I think you have all these references already in the introduction (where they are important). Here they can be omitted.
Response: deleted, thanks.

Eq. 2: I am not sure that this notation is elegant and fully appropriate. In my understanding, $m_p\_\theta$ is a function of $p$. Therefore $m\_\theta(p)$ would be, in my view, the more appropriate notation. The full equation should then read: $m\_\theta(p) = \theta_{skin} - \theta(p)$

Response: Our motivation for using a superscripted p was that it ensures unique symbols for each of the three variants of the conventional index (m_{\theta}, m_{\theta}^p, m_{\theta}^{p_crit}). However, we agree, the parentheses seem more intuitive, and one could also use the same symbol (m_{\theta}) and then denote the dependency on p in parentheses. To us it seems that the latter could have the disadvantage that the parentheses (p), or their absence, need to serve as the unique identifier in the text when referring to the different metrics via their symbol (e.g. m_\theta vs m_\theta(p)). Overall, we are also unsure about the optimal notation here. Both seem fine to us. To address the recommendation, we changed the notation as suggested, and checked that the use of the notation in the text allows unique identification. For consistency, we have also exchanged the use of p as subscripts to \theta by \theta(p) throughout.

L168: for the vertical levels → for all vertical levels
Response: adapted.

L183 and in many other places: during years → during the years
Response: adapted.

L209: should read "performance of the diagnostic …"
Response: adapted.

L214: should read "(in this case a diagnostic index)"
Response: adapted.

L219: confusing … what is $T$? Is $T = t_{end} - t_0$ ?
Response: yes; included the definition.

L222: should read "(a comparable approach … was used in Terpstra …"
Response: adapted.

Caption of Fig. 1: kg/kg → kg kg$^{-1}$
Response: adapted.

L257: m/s → m s$^{-1}$
Response: adapted.

L258: references should be in chronological order
Response: adapted.

L262: I don't understand "Resolved dynamics of cloud cover". But maybe it is anyway best to delete this sentence?
Response: deleted.

Caption of Fig. 2: please shorten this caption. The caption should only explain what the lines, colors, … in the plot mean, but the caption should not interpret the figures. The sentences "As expected …" and "The illustrated aspects …" can be deleted and the rest can be shortened
Response: shortened.

L290-300: here I had the feeling that this is mainly repetition from Sect. 2.3. Please consider to shorten this part.
Response: shortened.

Eq. 5: again the notation, I suggest to write $\theta(p^*)$ instead of the subscript.
Response: adapted.

L317: delete "defined here …", this was mentioned before.
Response: deleted.

L353: low or absent forcing ➔ weak or no forcing
Response: adapted.

L361: delete "or any fitting procedure" (or explain what it means)
Response: deleted.

L391: "as compared" ➔ "compared"
Response: adapted.

Eq. 7: and another notation: in the same spirit as above I suggest $m\_\theta(p_{crit}) = \theta_{skin} - \theta(p_{crit})$
Response: adapted.

L534: do you really want to emphasize the "slow-wind eye" here? In every circular symmetric vortex the wind speed must be zero at the center, so the "slow-wind eye" is not so much a surprise(?).
Response: we meant to emphasize that the spatial res. of ERA5 is high enough so that the 3D iso-surfaces computed in Met.3D neatly resolve this feature. Whilst not a surprise that there are slow winds in the center, we think it's an interesting perspective (in Fig. 3) to illustrate that the coherent volume of air with slow winds reaches high up into the stratosphere. However, apparently this didn't come across as desired in this part and it is certainly not central overall, so we have deleted that part of the sentence. Thanks.

L546: report ➔ reported
Response: adapted.

List of references: unlike the main paper, this part is not yet in very good shape. Often Page numbers, volume numbers or DOIs are missing. Sometimes journal names are abbreviated, sometimes not (please check in other WCD publications for the standard abbreviations to be used). Also please update the reference of the Afargan-Gerstman et al. paper.
Response: corrected, i.e.: added page numbers, volume number and DOIs were missing, consistent use of journal abbreviations, update of the Afargan-Gerstman et al. paper.